# MASTERING SPATIAL GRAPH PREDICTION OF ROAD NETWORKS

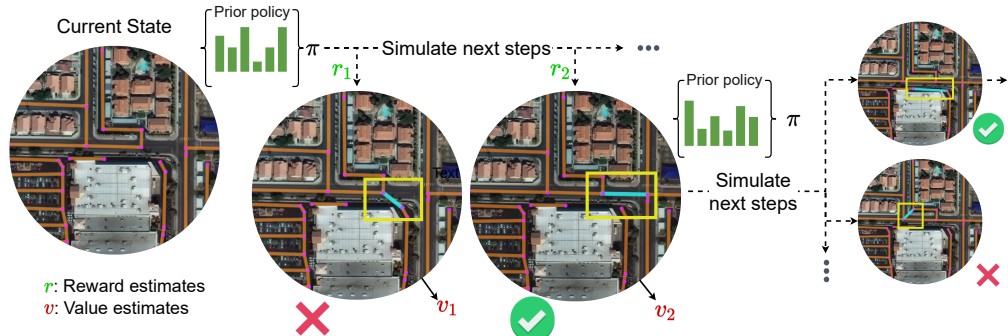

Figure 1: Our agent interacts with the currently generated spatial graph by proposing new edges to be added. A tree-based search produces a sequence of actions that maximizes a reward function based on complex geometrics priors.

## ABSTRACT

Accurately predicting road networks from satellite images requires a global understanding of the network topology. We propose to capture such high-level information by introducing a graph-based framework that simulates the addition of sequences of graph edges using a reinforcement learning (RL) approach. In particular, given a partially generated graph associated with a satellite image, an RL agent nominates modifications that maximize a cumulative reward. As opposed to standard supervised techniques that tend to be more restricted to commonly used surrogate losses, these rewards can be based on various complex, potentially non-continuous, metrics of interest. This yields more power and flexibility to encode problem-dependent knowledge. Empirical results on several benchmark datasets demonstrate enhanced performance and increased high-level reasoning about the graph topology when using a tree-based search. We further highlight the superiority of our approach under substantial occlusions by introducing a new synthetic benchmark dataset for this task.

## 1 INTRODUCTION

Road layout modelling from satellite images constitutes an important task of remote sensing, with numerous applications in and navigation. The vast amounts of data available from the commercialization of geospatial data, in addition to the need for accurately establishing the connectivity of roads in remote areas, have led to an increased interest in the precise representation of existing road networks. By nature, these applications require structured data types that provide efficient representations to encode geometry, in this case, graphs, a de facto choice in domains such as computer graphics, virtual reality, gaming, and the film industry. These structured-graph representations are also commonly used to label recent road network datasets (Van Etten et al., 2018) and map repositories (OpenStreetMap contributors, 2017). Based on these observations, we propose a new method for generating predictions directly as spatial graphs, allowing us to explicitly incorporate geometric constraints in the learning process, encouraging predictions that better capture higher-level dataset statistics.

In contrast, existing methods for road layout detection, mostly rely on pixel-based segmentation models that are trained on masks produced by rasterizing ground truth graphs. Performing pixel-wise segmentation, though, ignores structural features and geometric constraints inherent to the

problem. As a result, minimum differences in the pixel-level output domain can have significant consequences in the proposed graph, in terms of connectivity and path distances, as manifested by the often fragmented outputs obtained after running inference on these models. In order to address these significant drawbacks, we propose a new paradigm where we: (i) directly generate outputs as spatial graphs and (ii) formalize the problem as a game where we sequentially construct the output by adding edges between key points. These key points can in principle come from any off-the-shelf detector that identifies road pieces with sufficient accuracy. Our generation process avoids having to resort to cumbersome post-processing steps (Batra et al., 2019; Montoya-Zegarra et al., 2015) or optimize some surrogate objectives (Máttyus & Urtasun, 2018; Mosinska et al., 2018) whose relation to the desired qualities of the final prediction is disputed. Concurrently, the sequential decision-making strategy we propose enables us to focus interactively on different parts of the image, introducing the notion of a current state and producing reward estimates for a succession of actions. In essence, our method can be considered as a generalization of previous refinement techniques (Batra et al., 2019; Li et al., 2019b) with three major advantages: (i) removal of the requirement for greedy decoding, (ii) ability to attend globally to the current prediction and selectively target parts of the image, and (iii) capacity to train based on demanding task-specific metrics.

More precisely, our contributions are the following:

- We propose a novel generic strategy for training and inference in autoregressive models that removes the requirement of decoding according to a pre-defined order and refines initial sampling probabilities via a tree search.

- We create a new synthetic benchmark dataset of pixel-level accurate labels of overhead satellite images for the task of road network extraction. This gives us the ability to simulate complex scenarios with occluded regions, allowing us to demonstrate the improved robustness of our approach. We plan to release this dataset publicly.

- We confirm the wide applicability of our approach by improving the performance of existing methods on the popular SpaceNet and DeepGlobe datasets.

## 2 RELATED WORK

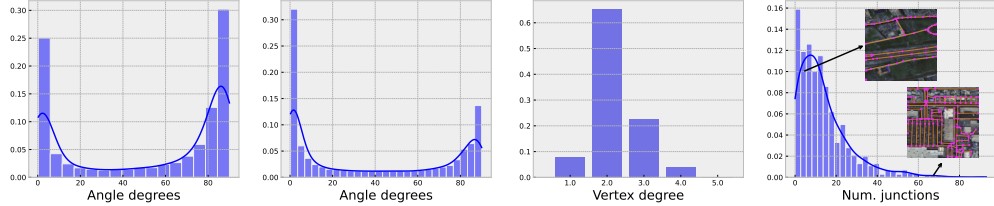

Figure 2: Typical road network features (SpaceNet dataset). From left to right: (a) the distribution of angles between road segments leading to the same intersection is biased towards 0 and 90 degrees, parallel or perpendicular roads. The same also holds (b) for angles between random road pieces within a ground distance of 400 meters. (c) Most road vertices belong to a single road piece, with a degree of 2. (d) The average number of intersections for areas of 400×400 meters by ground distance.

Initial attempts to extract road networks mainly revolved around handcrafted features and stochastic geometric models of roads (Barzohar & Cooper, 1996). Road layouts have specific characteristics, regarding radiometry and topology e.g. particular junction distribution, certain general orientation, and curvature (see Fig. 2), that enable their detection even in cases with significant occlusion and uncertainty (Hinz & Baumgartner, 2003). Modern approaches mostly formulate the road extraction task as a segmentation prediction task (Lian et al., 2020; Mattyus et al., 2015; Audebert et al., 2017) by applying models such as Hourglass (Newell et al., 2016) or LinkNet (Chaurasia & Culurciello, 2017). This interpretation has significant drawbacks when evaluated against structural losses, because of discontinuities in the predicted masks. Such shortcomings have been addressed by applying some additional post-processing steps, such as high-order conditional random fields (Niemeyer et al., 2011; Wegner et al., 2013) or by training additional models that refine these initial predictions (Máttyus et al., 2017; Batra et al., 2019). Other common techniques include the optimization of an ensemble of losses. Chu et al. (2019) rely on a directional loss and use non-maximal suppression as a thinning layer, while Batra et al. (2019) calculate orientations of road segments. Although such auxiliary losses somewhat improve the output consistency, the fundamental issue of producing

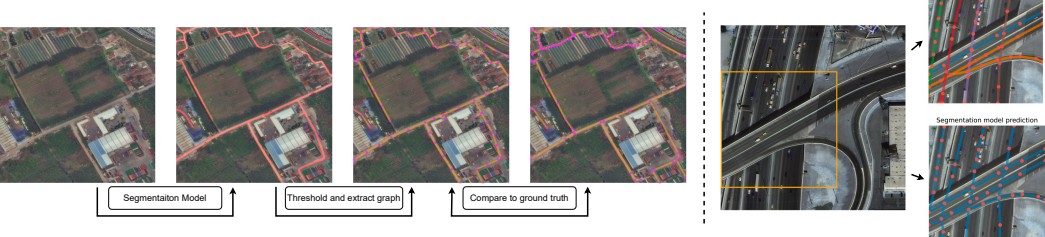

Figure 3: (Left) Typical segmentation-based methods generate an output graph by thresholding a segmentation mask, which can however often lead to fragmented outputs. Predicting segmentation masks also makes it impossible (right) to capture complex road interactions, such as overlapping roads at different elevations.

predictions in the pixel space persists. It remains impossible to overcome naturally occurring road network structures, e.g. crossings of roads in different elevations, see Fig. 3.

Previous failure cases have led to more intuitive conceptualizations of the task. Roadtracer (Bastani et al., 2018), iteratively builds a road network, similar to a depth-first search approach, while Chu et al. (2019) learn a generative model for road layouts and then apply it as a prior on top of a segmentation prediction mask. Proposed graph-based approaches, encode the road network directly as a graph, but either operate based on a constrained step-size (Tan et al., 2020) to generate new vertices or operate on a single step (He et al., 2020; Bandara et al., 2022), involving use-defined thresholding to post-process the final predictions. Most similar to our work, Li et al. (2019b) predict locations of key points and define a specific order traversing them, also similarly Xu et al. (2022). Such autoregressive models have been recently successfully applied with the use of transformers (Vaswani et al., 2017) in a range of applications (Nash et al., 2020; Para et al., 2021a;b; Xu et al., 2022) to model constraints between elements, while their supervised training explicitly requires tokens to be processed in a specific order. This specific order combined with the fact that only a surrogate training objective is used, introduces limitations, discussed further in the next section. In order to eliminate this order requirement and to optimize based on the desired metric, while attending globally to the currently generated graph, we propose to use RL as a suitable alternative.

When generating discrete outputs, an unordered set of edges (Zaheer et al., 2017), it is challenging to adapt existing learning frameworks to train generative models (Para et al., 2021b). Instead of optimizing in the image space, however, we are interested in optimizing spatial structured losses by learning program heuristics, i.e. policies. RL has found success in the past in computer vision applications (Le et al., 2021), but mainly as an auxiliary unit with the goal of improving efficiency (Xu et al., 2021) or as a fine-tuning step (Qin et al., 2018). We instead rely on RL to produce the entire graph exploiting the ability of the framework for more high-level reasoning.

## 3 METHODOLOGY

We parametrize a road network as a graph $\mathcal{G} = \{\mathcal{V}, \mathcal{E}\}$ with each vertex $v_i = [x_i, y_i]^\top \in \mathcal{V}$ representing a key point on the road surface. The set of edges $(v_i, v_j) \in \mathcal{E}$, corresponds to road segments connecting these key points. We can then generate a probability distribution over roads by following a two-step process: i) generation of a set of vertices and ii) generation of a set of edges connecting them. Formally, for an image $\mathcal{I}$, a road network $\mathcal{R}$ is derived as:

$$\mathcal{R} = \arg\max_{\mathcal{V}, \mathcal{E}} P(\mathcal{V}, \mathcal{E} \mid \mathcal{I}) = P(\mathcal{E} \mid \mathcal{V}, \mathcal{I})P(\mathcal{V} \mid \mathcal{I}). \tag{1}$$

The graph nodes typically correspond to local information in an image, and we therefore resort to a CNN-based model to extract key points, providing the set $\mathcal{V}'$, that sufficiently captures the information in the ground truth graph $\mathcal{G}$. The construction of edges, however, requires higher-level reasoning that can cope with parallel roads, junctions, occlusions, or poor image resolution, among other difficulties.

Considering probabilistic models over sequences and using the chain rule, we can factorize the joint distribution as the product of a series of conditional distributions

$$P(\mathcal{E} \mid \mathcal{V}, \mathcal{I}; \sigma) = \prod_{n=1}^{N_{\mathcal{E}}} P(e_{\sigma(n)} \mid e_{<\sigma(n)}, \mathcal{V}, \mathcal{I}), \tag{2}$$

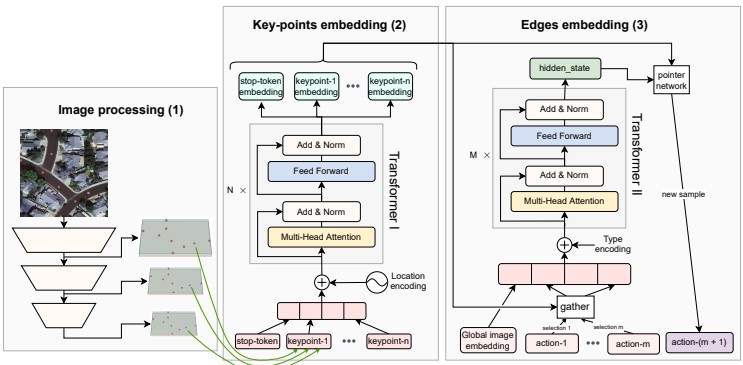

Figure 4: The base autoregressive model with its three main components. (1) A backbone image model (ResNet) that extracts features for each key point at different scales, along with a global image embedding. (2) A key point model embeds visual and location features of distinct key points. (3) An edge embedding model relates the current edge sequence with the respective key points. Each edge token (signalled with the tokens 'action-$i$') corresponds to an index specifying the respective key point. A pair of such tokens designates an edge as its two endpoints. At the end of (3) we obtain a new distribution over key points, that leads to an incremental update to the graph, after sampling.

where $e_{<\sigma(n)}$ represents $e_{\sigma(1)}, e_{\sigma(2)}, \ldots, e_{\sigma(n-1)}$ and $\sigma \in S_{N_{\mathcal{E}}}$ denotes the set of all permutations of the integers $1, 2, \ldots, N_{\mathcal{E}}$, with $N_{\mathcal{E}}$ the number of edges. For our work, we consider the setting where these sequences are upper bounded in length, i.e. $N_{\mathcal{E}} \leq N_{\max}$, a reasonable assumption when dealing with satellite images of fixed size. Autoregressive models (ARMs) have been used to solve similar tasks in the past by defining a fixed order of decoding (Oord et al., 2016; van den Oord et al., 2016; Nash et al., 2020; Para et al., 2021a). In our case, this would correspond to sorting all key points by their $x$ and $y$ locations and generating edges for each of them consecutively. We call this the *autoregressive order*. There are, however, two major drawbacks.

First, the evaluation metrics used for this task define a buffer region in which nodes in the ground truth and the predicted graph are considered to be a match. Therefore, a newly generated edge can be only partially correct, when only partially overlapping with the ground truth graph. This non-smooth feedback comes in clear contrast to the supervised training scheme of ARMs, minimization of the negative log-likelihood, that assumes perfect information regarding the key points' locations, i.e. that the sets $\mathcal{V}$ and $\mathcal{V}'$ are the same. In practice, this condition is rarely met, as the exact spatial graph can be represented in arbitrarily many ways by subdividing long edges into smaller ones or due to small perturbation to key points' locations. It is thus imperative that our model can estimate the expected improvement of adding selected edges, which implicitly can also signal when to appropriately end the generation process.

Second, the requirement to decode according to the autoregressive order introduces a bias and limits the expressiveness of the model (Uria et al., 2014). As a result, it can lead to failures in cases with blurry inputs or occlusions (Li et al., 2019b). Previous solutions include the use of beam search, either deterministic or stochastic (Meister et al., 2021). Beam search does not however eliminate the bias introduced in the selection order of the key points, while suffering from other deficiencies, such as degenerate repetitions (Holtzman et al., 2019; Fan et al., 2018). In order to address these shortcomings, we advocate for a permutation invariant strategy. We present a novel generic strategy, which improves autoregressive models without requiring significantly more computational cost.

## 3.1 AUTOREGRESSIVE MODEL

We start by introducing a base autoregressive model, illustrated in Fig. 4. Given an image and a set of key points, our model produces a graph by sequentially predicting a list of indices, corresponding to the graph's flattened, unweighted edge-list. Each forward pass produces probabilities over the set of key points, which leads to a new action after sampling. A successive pair of indices defines an edge as its two endpoints. A special end-of-sequence token is reserved to designate the end of the generation process.

Following Wang et al. (2018); Smith et al. (2019), we begin by extracting visual features per key point, by interpolating intermediate layers of a ResNet backbone to the key points' locations, which

are further augmented by position encodings of their locations. We then further process these features using two lightweight Transformer modules. The first transformer (Transformer I in Fig. 4) encodes the features of the key points as embeddings. The second transformer (Transformer II in Fig. 4) takes as input the currently generated edge list sequence, corresponding to the currently partially generated graph. Edges are directly mapped to the embeddings of their comprising key points, supplemented by position and type embeddings, to differentiate between them, as shown in Fig. 5 (a). An additional global image embedding, also extracted by the ResNet, is used to initialize the sequence. The Transformer II module produces a single hidden state, which is linked with the $N_{\mathcal{V}'} + 1$ (corresponding to the provided key points, supplemented by the special end of the generation token) key points' embeddings by a pointer network (Vinyals et al., 2015), via a dot-product to generate the final distribution. This allows a variable number of actions that depends on the current environment state, instead of using a fixed action space.

## 3.2 AUGMENTED SEARCH

In order to address the problems of greedy decoding (analysed Section 3), we frame our road extraction task as a classical Markov-decision process (MDP). The generation of a graph for every image defines an environment, where the length of the currently generated edge list determines the current step. Let $\boldsymbol{o}_t$, $\alpha_t$ and $r_t$ correspond to the observation, the action and the observed reward respectively, at time step $t$. The aim is to search for a policy that maximizes the expected cumulative reward over a horizon $T$, i.e., $\max_\pi J(\pi) \coloneqq \mathbb{E}_\pi[\sum_{t=0}^{T-1} \gamma^t r_t]$ where $\gamma \in (0, 1]$ indicates the discount factor and the expectation is with respect to the randomness in the policy and the transition dynamics. We set the discount factor to 1 due to the assumed bounded time horizon, and we note that although the dynamics of the environment are deterministic, optimizing the reward remains challenging.

Each action leads to the selection of a new key point, with new edges being added once every two actions. The addition of a new edge leads to a revision of the predicted graph and triggers an intermediate reward

$$r_t = \mathrm{sc}(\mathcal{G}_{\mathrm{gt}}, \mathcal{G}_{\mathrm{pred}_t}) - \mathrm{sc}(\mathcal{G}_{\mathrm{gt}}, \mathcal{G}_{\mathrm{pred}_{t-1}}), \tag{3}$$

where $\mathrm{sc}(\mathcal{G}_{\mathrm{gt}}, \mathcal{G}_{\mathrm{pred}_t})$ is a similarity score between the ground truth graph $\mathcal{G}_{\mathrm{gt}}$ and the current estimate $\mathcal{G}_{\mathrm{pred}_t}$. Discussion of the specific similarity scores used in practice is postponed for Section 3.3.

A proper spatial graph generation entails (i) correct topology and (ii) accurate location prediction of individual roads. For the latter, intermediate vertices of degree 2 are essential. We call a road segment (RS), an ordered collection of edges, between vertices of degree $d(.)$ two (or a collection of edges forming a circle):

$$\mathrm{RS} = \{(\boldsymbol{v}_{\mathrm{rs}_1}, \boldsymbol{v}_{\mathrm{rs}_2}), \dots, (\boldsymbol{v}_{\mathrm{rs}_{k-1}}, \boldsymbol{v}_{\mathrm{rs}_k})\} \text{ s.t } (\boldsymbol{v}_{\mathrm{rs}_i}, \boldsymbol{v}_{\mathrm{rs}_{i+1}}) \in \mathcal{E} \text{ for } i = 1, \dots, k-1$$
$$d(\boldsymbol{v}_{\mathrm{rs}_i}) = 2, \text{ for } i = 2, \dots k-1, \quad (d(\boldsymbol{v}_{\mathrm{rs}_1}) \neq 2 \text{ and } d(\boldsymbol{v}_{\mathrm{rs}_k}) \neq 2 \text{ or } \boldsymbol{v}_{\mathrm{rs}_1} = \boldsymbol{v}_{\mathrm{rs}_k}).$$

During the progression of an episode (i.e. the sequential generation of a graph), the topological nature of the similarity scores in Eq. 3 implies that the effect of each new edge to the reward will be reflected mostly once its whole corresponding road segment has been generated. To resolve the ambiguity in the credit assignment and allow our agent to look ahead into sequences of actions, we rely on Monte Carlo Tree Search (MCTS) to simulate entire sequences of actions. We use a state-of-the-art search-based agent, MuZero (Schrittwieser et al., 2020), that constructs a learnable model of the environment dynamics, simulating transitions in this latent representation and leading to significant computational benefits.

Specifically, MuZero requires three distinct parts (see also Fig. 5):

1. A representation function $f$ that creates a latent vector of the current state $\boldsymbol{h}_t = f_\theta(\boldsymbol{o}_t)$. For this step, we use the autoregressive model, as shown in Fig. 4. Our current latent representation $\boldsymbol{h}_t$ contains the graph's hidden state, along with the key points' embeddings used to map actions to latent vectors. As key points remain the same throughout the episode, image-based features (Components (1) and (2) in Fig. 4) are only computed once.

2. A dynamics network $g$, we use a simple LSTM (Hochreiter & Schmidhuber, 1997), that predicts the effect of a new action by predicting the next hidden state and the expected reward: $(\hat{\boldsymbol{h}}_t, \hat{r}_t) = g_\theta(\tilde{\boldsymbol{h}}_{t-1}, \alpha_t)$. We can replace $\tilde{\boldsymbol{h}}_{t-1}$ with the latent representation $\boldsymbol{h}_{t-1}$, or its previous computed approximation $\hat{\boldsymbol{h}}_{t-1}$ for tree search of larger depth larger than 1.

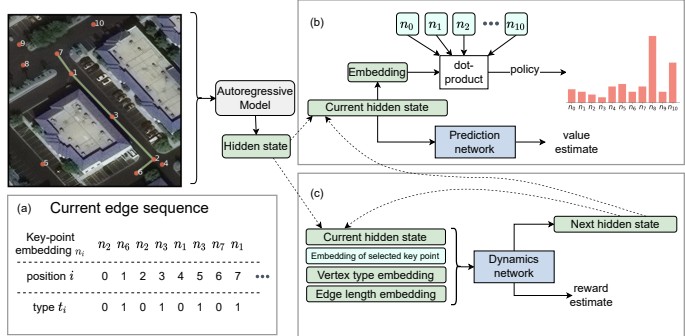

Figure 5: The autoregressive model generates a hidden state corresponding to the graph embedding. (a) When doing so, the edge decoder directly attends to the input key points' embeddings, augmented by position and type embeddings. (b) The prediction network uses the hidden state to produce value and policy predictions. A pointer network allows an intuitive scale of the action space by the number of key points. (c) The dynamics network simulates trajectories by estimating new hidden states and rewards. For newly generated edges, it takes as input the embeddings of the new key points, but also the degree of the two vertices involved and the length of the newly proposed generated edge.

3. A prediction network $\psi$, that estimates the policy and the value for the current state $(\boldsymbol{p}_{t+1}, v_t) = \psi_\theta(\tilde{\boldsymbol{h}}_t)$. We compute the policy via a pointer network, as described in Section 3.1. Value estimates are produced by a simple multi-layer network.

The dynamics network guides the search and evaluates the expected reward of actions. For every newly generated edge, we also explicitly inform the network regarding the creation of new intersections and the expected relative change in the overall road surface generated via embeddings (see Fig. 5). By using the dynamics network, we bypass the expensive call to the decoder module during the search, and can instead approximate small modifications in the latent representation directly. For our experiments, the dynamics network requires up to 90 times less floating-point operations to simulate trajectories, compared to using the edge embeddings' decoder. Effectively, our method does not involve significantly more computation budget compared to the base autoregressive model.

### 3.3 EVALUATION METRICS

We adopt the same evaluation metrics both as a comparison between different methods but also as the incremental rewards for our agent, by Eq. 3. We use the relaxed versions of precision, recall and intersection over union for pixel-level predictions *Correctness/Completeness/Quality* (CCQ) (Wiedemann et al., 1998; Wang et al., 2016). As graph-theoretic metrics we use *APLS* (Van Etten et al., 2018) and additionally include new metrics introduced in Citraro et al. (2020) that compare *Paths*, *Junctions* and *Sub-graphs* of the graphs in question, producing respectively *precision*, *recall* and $f_1$ scores. More details can be found in Appendix E.1

## 4 EXPERIMENTS

**Implementation details** We resize images to $300 \times 300$ pixels, standardizing according to the training set statistics. For exploration, we initialize workers using Ray (Moritz et al., 2017) that execute episodes in the environment. For training, we unroll the dynamics function for $t_d = 5$ steps and use priority weights for the episode according to the differences between predicted and target values. Our algorithm can be considered as an approximate on-policy TD($\lambda$) (Sutton & Barto, 2018) due to the relatively small replay buffer. We reanalyse older games (Schrittwieser et al., 2020) to provide fresher target estimates. Unvisited graph nodes are selected based on an upper confidence score, balancing exploration with exploitation, similar to Silver et al. (2018). We add exploration noise as Dirichlet noise and select actions based on a temperature-controlled sampling procedure, whose temperature is reduced during training.

Given the limited high-quality available ground truth labels (Singh et al., 2018) and to accelerate training, we employ modifications introduced in EfficientZero (Ye et al., 2021). We investigate adding supervision to the environment model and better initialize Q-value predictions similar to the implementation of Elf OpenGo (Tian et al., 2019). We further scale values and rewards using an

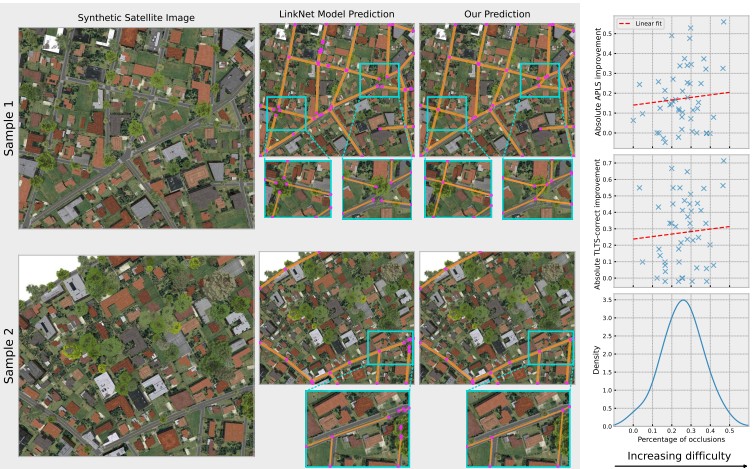

Figure 6: (Left) Examples of fragmented outputs from the LinkNet model under cases of significant occlusion and how our approach performs in these demanding circumstances. (Right) The performance gap between our method and the same baseline is wide, based on topological spatial graph metrics, for a wide range of difficulties in the images. More details and examples are given in the supplementary material.

invertible transform inspired by Pohlen et al. (2018). Here, we predict support, as fully connected networks are biased towards learning low-frequency representations (Jacot et al., 2018). Selecting new actions involves generating simulations that can be done expeditiously given the small dimension of the latent space and the modest size of the dynamics network. Finally, to generate key points, we skeletonize segmentation masks provided by any baseline segmentation model, by thresholding the respective segmentation masks produced and applying RDP-simplification (Douglas & Peucker, 1973; Ramer, 1972). Selecting an appropriate threshold and subdividing larger edges guarantees that the generated set $\mathcal{V}'$ adequately captures most of the ground truth road network, leaving the complexity of the problem for our model to handle.

### 4.1 SYNTHETIC DATASET

We generate a dataset of overhead satellite images of a synthetic town using CityEngine[1]. We randomly specify vegetation of varying height and width along the side walks of the generated streets, leading inadvertently to occlusions of varying difficulty. The simulated environment allows specifying *pixel-perfect masks* regarding both *roads and trees* occluding the road surface based on the provided camera parameters (Kong et al., 2020). We can hence tune the complexity of the task and quantify the benefits of our approach for varying levels of difficulty. We defer more details regarding the generation process and dataset examples to the supplementary material.

We compare our method by training on our dataset a *LinkNet* model (Chaurasia & Culurciello, 2017), a popular segmentation model that has been widely used in the remote sensing community (Li et al., 2019a). Even in this synthetic and thus less diverse scenario, the deficiency of segmentation models to rely mostly on local information, with no explicit ability for longer-range interactions, is evident. Fig. 6, illustrates examples of such over-segmented predictions and how our approach can improve on them. We also define a 'difficulty' attribute per synthetic satellite image, quantifying the occlusions as a percentage of the ground truth road mask covered. We observe a considerable absolute improvement in topological metric scores when training our model on this synthetic dataset, compared to the LinkNet baseline, for varying image difficulty.

### 4.2 REAL DATASETS

We evaluate our method on the SpaceNet and DeepGlobe datasets. We use the same train-test splits as in Batra et al. (2019) to promote reproducibility, while results are reported for the final combined graph on the original image scale. No pre-training on the synthetic dataset takes place. Further details regarding pre-processing are available in the Appendix E.2

---

[1]https://www.esri.com/en-us/arcgis/products/arcgis-cityengine/overview

Table 1: Quantitative results for the SpaceNet and DeepGlobe datasets.

| | Method | CCQ | | | TLTS | | APLS ↑ | Path-Based | | | Junction-Based | | | Sub-graph-Based |
|---|---|---|---|---|---|---|---|---|---|---|---|---|---|---|
| | | corr. ↑ | comp. ↑ | qual. ↑ | corr. ↑ | 2l+2s ↓ | | pre. ↑ | rec. ↑ | f1 ↑ | pre. ↑ | rec. ↑ | f1 ↑ | f1 ↑ |
| SpaceNet | DeepRoadMapper (Máttyus et al., 2017) | 0.6943 | 0.6838 | 0.5386 | 0.4110 | 0.1012 | 0.5143 | 0.5958 | 0.6400 | 0.6171 | 0.6293 | 0.7443 | 0.6820 | 0.6783 |
| | Segmentation (Long et al., 2015; Kaiser et al., 2017) | 0.7493 | 0.7094 | 0.5969 | 0.4143 | 0.0828 | 0.5454 | 0.6909 | 0.6863 | 0.6885 | 0.7186 | 0.7710 | 0.7438 | 0.7117 |
| | LinkNet (Chaurasia & Culurciello, 2017) | 0.8100 | 0.7449 | 0.6409 | 0.4894 | 0.0743 | 0.5743 | 0.6719 | 0.6460 | 0.6586 | 0.6985 | 0.7809 | 0.7374 | 0.7576 |
| | Orientation (Batra et al., 2019) | 0.8070 | 0.8001 | 0.6862 | 0.5594 | 0.0884 | 0.6315 | 0.7175 | 0.7280 | 0.7227 | 0.7552 | 0.7591 | 0.7571 | 0.7802 |
| | Sat2Graph (He et al., 2020)** | 0.6917 | 0.7351 | 0.5734 | 0.5802 | 0.1104 | 0.5951 | 0.5952 | 0.5416 | 0.5671 | 0.7474 | 0.5951 | 0.6626 | 0.7180 |
| | SPIN road mapper (Bandara et al., 2022) | 0.7837 | 0.7988 | 0.6621 | 0.5922 | 0.1058 | 0.6422 | 0.7276 | 0.7265 | 0.7270 | 0.7621 | 0.7827 | 0.7722 | 0.7837 |
| | Ours | 0.7726 | 0.7852 | 0.6632 | 0.5970 | 0.0878 | 0.6523 | 0.7323 | 0.7543 | 0.7431 | 0.7762 | 0.7824 | 0.7792 | 0.7922 |
| Deep Globe | LinkNet (Chaurasia & Culurciello, 2017) | 0.8012 | 0.8676 | 0.7328 | 0.6640 | 0.0804 | 0.6525 | 0.6882 | 0.6920 | 0.6901 | 0.7675 | 0.7444 | 0.7558 | 0.7879 |
| | Orientation (Batra et al., 2019) | 0.8243 | 0.8857 | 0.7545 | 0.6866 | 0.1047 | 0.7012 | 0.6937 | 0.8082 | 0.7465 | 0.7624 | 0.7939 | 0.7778 | 0.8282 |
| | Ours* | 0.8163 | 0.8929 | 0.7391 | 0.7177 | 0.1058 | 0.7398 | 0.7050 | 0.8181 | 0.7573 | 0.7834 | 0.8253 | 0.8038 | 0.8338 |

* We do not fine-tune our model on the DeepGlobe dataset but instead refine predictions standardizing according to train dataset statistics.
** The authors provided predictions corresponding only to a center crop of the original SpaceNet dataset images. Also note that the test set is different from the one reported on the rest of the methods, see also Appendix.
Blue: best score, Green: second best score, Gray: results reported in different test set

### 4.2.1 COMPARISON TO BASELINES

We first verify that under the effectiveness of the proposed approach under an ideal scenario where the key points conditioned upon, correspond to the ones from the ground truth. In the interest of space we point the reader to Appendix A and Table 3. Subsequently, we move to the primary task of predicting spatial graphs without the ground-truth graph information but extract key points via the aforementioned process and train using the described topological metrics directly. The previous baselines are not applicable in this case, due to lack of ground truth information, so we instead compare against the following; we explore powerful CNN architectures, by training a *Segmentation* model with a ResNet backbone. We evaluate *DeepRoadMapper* (Máttyus et al., 2017), a model that refines previous segmentation by adding connections along possible identified paths. As done by Batra et al. (2019) we notice that in complex scenarios, the effect of this post-processing step is not always positive. We also evaluate against *LinkNet* (Chaurasia & Culurciello, 2017), and *Orientation* (Batra et al., 2019), which is trained to predict simultaneously road surfaces and orientation.

Table 2: Qualitative results of improved connectivity. We recommend zooming in for more details

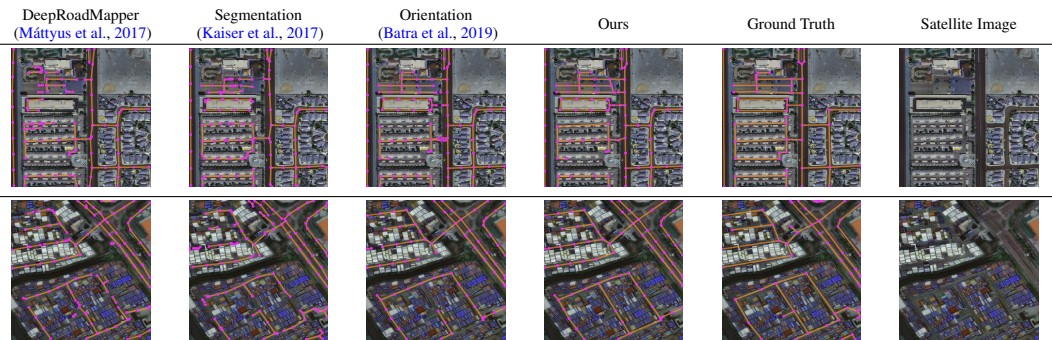

Quantitative results in Table 1 and visual inspection in Table 2, affirm that the global context and the gradual generation incite a better understanding of the scene, leading to consistently outperforming topological metric results compared to the baselines. We remark that our predictions are more topologically consistent with fewer shortcomings, such as double roads, fragmented roads, and over-connections. This is further supplemented by comparing the statistics of the predicted spatial graphs in Fig. 7. We further showcase the transferability of our model by employing it with no fine-tuning (apart from dataset-specific image normalization) on the DeepGlobe dataset. We can refine previous predictions by adding missing edges, leading to more accurate spatial graph predictions, as shown in Table 1. This confirms our conjecture that road structures and geometric patterns are repeated across diverse cities' locations.

### 4.2.2 ABLATION STUDY

We experimented attending to image features for the two transformer modules by extracting per-patch visual features from the conditioning image $H^{\text{img}} = [\boldsymbol{h}_1^{\text{img}}, \boldsymbol{h}_2^{\text{img}}, \ldots]$, as done in the Vision Transformer (Dosovitskiy et al., 2020). This did not lead to significant improvements, which we attribute to over-fitting. In Fig. 8 we highlight the relative importance of some additional components for the final predictions. As efficiency is also of particular importance to us, we further visualize the effect of varying the simulation depth of the dynamics network during training. Surprisingly

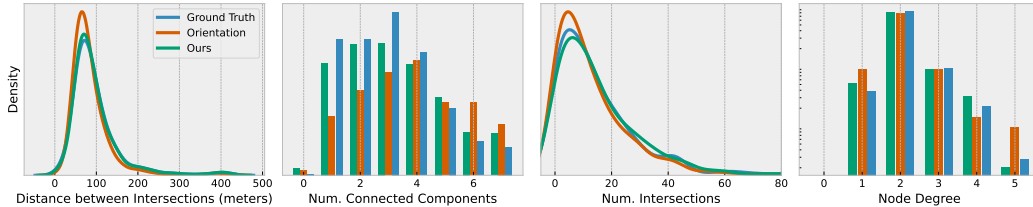

Figure 7: Comparison of generated graph statistics, following the same post-processing, averaged across regions of 400×400 meters in ground distance. Orientation refers to the method of Batra et al. (2019).

| Model | APLS | P-f1 | J-f1 | S-f1 |
|---|---|---|---|---|
| Ours | **0.6523** | **0.7431** | **0.7792** | **0.7922** |
| − autoregressive pretraining | -15.3% | -13.7% | -14.1% | -13.3% |
| − visual features for key-points | -13.4% | -12.7% | -12.1% | -12.3% |
| − tree-search during evaluation | -2.1% | -1.4% | -1.7% | -1.1% |
| + cross attend to image features | +0.2% | -0.4% | -0.7% | -0.3% |

P: Path-based, J: Junction-based, S: Sub-graph-based

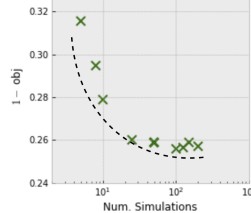

Figure 8: (Left) Ablation study on the SpaceNet dataset. (Right) The estimated Pareto front achieved by relating the number of simulations during training and the objective obj = avg(APLS + Path-based f1 + Junction-based f1 + Sub-graph-based f1) for the same time budget.

perhaps, our method performs consistently better than baselines, even for a small overall simulation length, as this already enables better policy approximations.

In Appendix A we provide incremental results for the task of predicting road networks based on a optimal set of key points. In Appendix B we provide insights concerning interpretability and further comparison to baselines based on the varying difficulty of the predicted underlying road networks. In Appendix C we give more information regarding the generation of the synthetic dataset, while in Appendix D more information regarding the model architecture. Final in Appendix E we provide more implementation decisions, including details on exactly how key points and generated and how individual patch-level predictions are fused together. More examples of full environment trajectories are given in Appendix F. We stress that our method can act on partially initialized predictions, registering it also as a practical refinement approach on top of any baseline. Initializing our model according to the ARM model allows a moderately quick fine-tuning phrase. In combination with the learned environment model, which circumvents expensive calls to the edge embedding model for each simulation step in the MCTS, allows us to train even on a single GPU.

## 5 CONCLUSIONS

We presented a novel reinforcement learning framework for generating a graph as a variable-length edge sequence, where a structured-aware decoder selects new edges by simulating action sequences into the future. Importantly, this allows the model to better capture the geometry of the targeted objects. This is confirmed by our experimental results, since our approach consistently produces more faithful graph statistics. One advantage of the proposed method is that the reward function is based on (non-continuous) metrics that are directly connected to the application in question. Our approach does not require significantly more computational resources compared to state-of-the-art supervised approaches, while in addition, it can be used to refine predictions from another given model. We also remark that the direct prediction of a graph enables the concurrent prediction of meta-information about the edges, including, for instance, the type of road (highway, primary or secondary street, biking lane, etc).

Our approach opens the door to several directions for future work. For example, we have assumed that a pre-defined model gives the location of key points, but one could instead augment the action space to propose new key points' locations. Other promising directions include the direct prediction of input-dependent graph primitives, e.g. T-junctions or roundabouts. Finally, we emphasize that our approach is suitable to a wide variety of applications where autoregressive models are typically used, and it is of special interest when there is a need for complex interactions or constraints between different parts of an object.

## 6 REPRODUCIBILITY STATEMENT

We have taken multiple steps to ensure reproducibility of the experiments. We refer the reader to Appendix E for a complete description of the training protocol. We have also released the code as part of the supplementary material, including scripts on how to reproduce our results.

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

Table 3: APLS metric and perplexity (bits of information per edge) results for the SpaceNet dataset with provided key points that adequately capture the ground truth graph. Autoregressive order is defined in Section 3, while random order entails random permutation between edges and the order of key points within edges. To calculate perplexity for our method we use the initial predicted policy distribution, without any additional search.

| Metric \ Method | Random | Cls | GCN | ARM | Ours |
|---|---|---|---|---|---|
| APLS | 0.008 | 0.310 | 0.488 | 0.743 | **0.778** |
| Bits per edge: Autoregressive order | 8.743 | - | - | **0.733** | 4.636 |
| Bits per edge: Random order | 8.743 | - | - | 30.84 | **4.565** |

## A  MORE EXPERIMENTS

We first assess the performance of our proposed method in an ideal scenario where the key points, correspond to the ones from the ground truth. To hinder training and inference, we insert additional key points as (1) random intermediate points between known edges and (2) randomly sampled locations in the images. Here, our assumption in Section 3 that the set $\mathcal{V}'$ suffices to generate the ground truth graph, holds by construction. We compare our method against several baselines that learn to connect edges between key points, using the same feature extraction pipeline, described in Section 3.1, as our model. *Cls* is a classification network that predicts for all pairs of key points a value $\{0, 1\}$ corresponding to the existence of an edge. *GCN* implements a graph neural network that predicts directly the adjacency matrix. We also present an autoregressive version of our model *ARM*, that is trained with cross-entropy loss to predict the pre-defined ordered sequence of key points. We use this model to initialize ours. Results are presented in Table 3.

As expected, the ARM model achieves a low perplexity score when evaluated against the corresponding sequence, ordered according to the autoregressive order, but suffers in predicting the edges when in random order. The ARM underperforms because of frequent early terminations and the implicit inability to revisit key points, what the desired final metric is concerned, here APLS. Even though our model is developed upon this autoregressive model, it generates tokens in an arbitrary arrangement. Reward and value estimates enable a different training scheme that deeply correlates with the desired objective.

## B  INTERPRETABILITY

We visualize attention (of the Transformer II module), using the attention flow proposed in Abnar & Zuidema (2020), in Fig. 9. To create attention scores per edge, we aggregate scores for the pair of tokens that define each edge. New predictions lay increased attention to already generated junctions, parallel road segments, and other edges belonging to the same road segment.

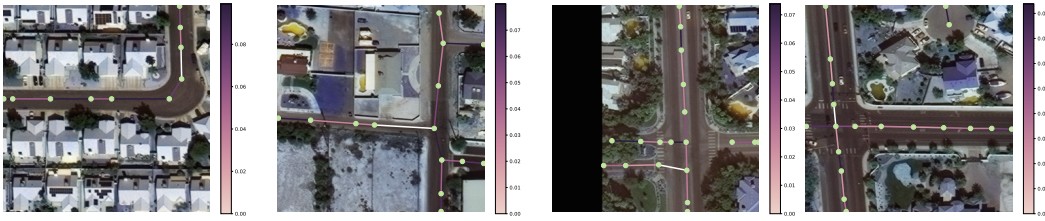

Figure 9: We visualize attention by aggregating attention scores for key points that form the same edge. White denotes the latest edge added, for which the attention scores are calculated. Colours indicate the amount of attention on any current edge in the graph.

We also compare APLS results achieved by varying the difficulty of the ground truth images in terms of the total number of junctions (vertices with a degree greater than 2) and in terms of the average length of road segments that are present, in Fig. 10. Our method explicitly captures information re-

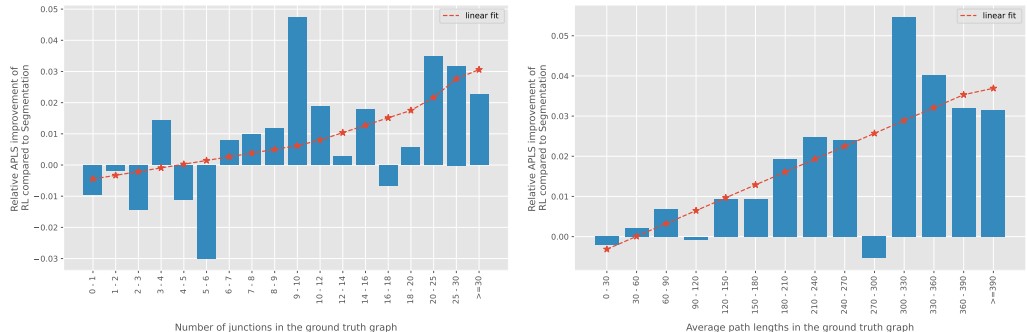

Figure 10: APLS improvement of our method, compared to the Orientation method Batra et al. (2019), based on images capturing a ground distance area of $400 \times 400$ meters of the SpaceNet dataset. (Left) We vary the number of junctions present, and (right) the average road segment length (in meters).

garding the degree of the key points during the search, while it can encode better global information, even across larger distances. It is not a surprise perhaps then, that it outperforms the baselines more convincingly as the difficulty of the ground truth road network increases.

Finally, we visualize an example of an imagined rollout trajectory at a single step of our algorithm in Fig. 11. During a single inference step, our method uses tree search to look ahead into sequences of actions in the future. For our example, we have chosen a relatively smaller number of simulations (10) for better visual inspection. We also show the corresponding environment states reached, which are, however, not explicitly available to the model, as it is searching and planning using a learned model of the environment.

## C  DATASET CREATION

We use CityEngine, a 3D modelling software for creating immersive urban environments. We generate a simple road network and apply a rural city texture on the created city blocks, provided by Kong et al. (2020). We then uniformly generate trees of varying height and size along the side walks of the generated streets. We then iteratively scan the generated city by passing a camera of specific orientation and height. We repeat the same process after suitable modifications to the texture, for the generation of the street masks, as well as the vegetation masks, that correspond to only the plants along the side walks. Some examples of the generated images are provided in Fig. 12. We note that additional occlusion can be caused by the relation of the camera with the 3D meshes corresponding to buildings. These occlusions are, however, not captured by our generated masks, and we can expect them to contribute partially to the fragmented segmentation results.

We train a segmentation-based model, LinkNet, as our baseline. We rasterize the ground truth graph to create pixel-level labels and train by maximizing the intersection over union, which is commonly done in practice. We note that there is a tradeoff between the nature of the predictions and the choice of the line-width with which the ground truth graph is rasterized. A large width achieves better results in terms of connectivity of the predicted graph but results in poorer accuracy in the final key points' locations. Furthermore, when providing a large width, areas in the image with more uncertainty, e.g. vegetation that is not above a road segment, are also predicted as road networks with high certainty, leading to spurious, disconnected road segments. To highlight the advantages of our method compared to this baseline and in order to promote more meaningful predictions, we select a relatively smaller width.

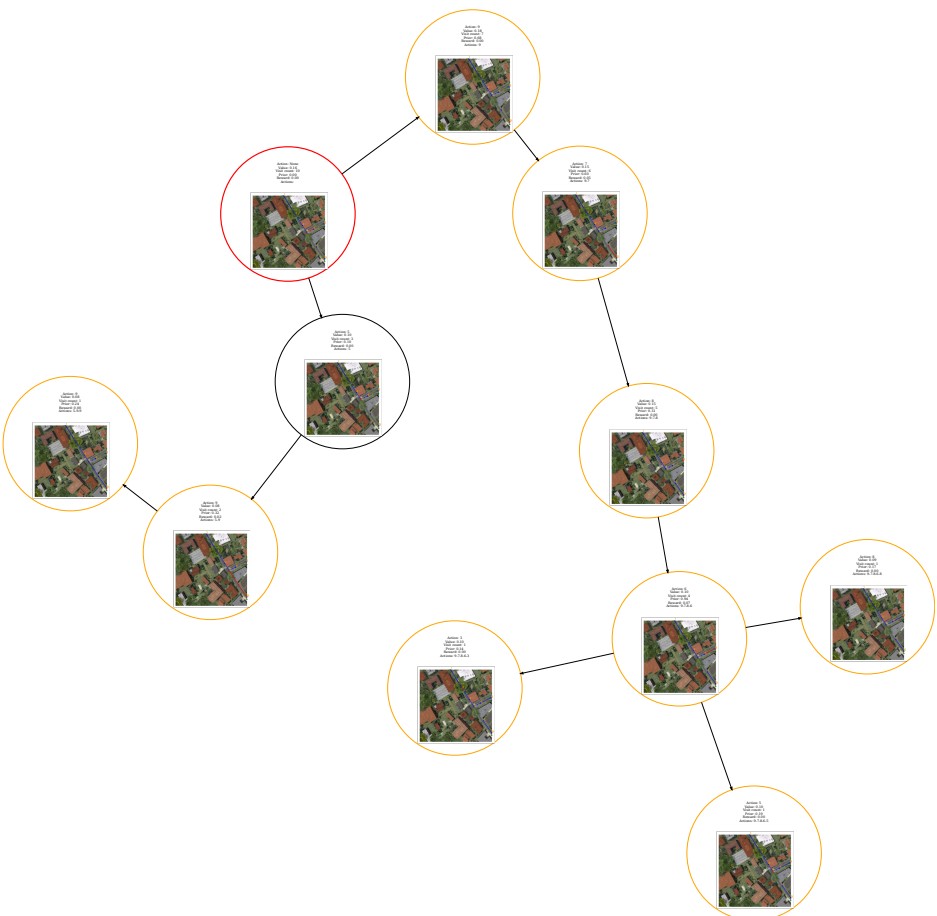

Figure 11: Example of a "dreamt" tree search of our model (we recommend zooming in for more details). Here, images correspond to the environment observations by following the respective sequence of actions, which are, however, not given as input to the model but only portrayed for visual purposes. The model has only access to the original observation and "dreamt" trajectories on the learned latent space. The root of the search tree is indicated by the colour red. Orange nodes correspond to the children that attain the highest estimated value. We also provide reward and value estimates of our model based on the current latent representation. We perform a smaller number of simulations into the future, for visual purposes.

## D  ARCHITECTURE DETAILS

As an image backbone model, we use a ResNet-18 for the synthetic dataset and a ResNet-50 for the real dataset experiments. We extract features at four different scales, after each of the 4 ResNet layers. To extract features for each key point, we interpolate the backbone feature maps based on the key points' locations. We use different learned embeddings based on the actual key points' locations. For the key points embedding model, we use a transformer encoder with 16 self-attention layers and a dropout rate of 0.15. We use layer normalization and GELU activation functions.

For the edge-embeddings model, we use the respective key points embedding, along with learned position and type embeddings, which we all sum together. As aforementioned, we can initialize the current edge sequence based on previous predictions, allowing our model to refine any initial prediction provided. Again, we use the same transformer architecture with 16 self-attention layers, and a dropout rate of 0.15.

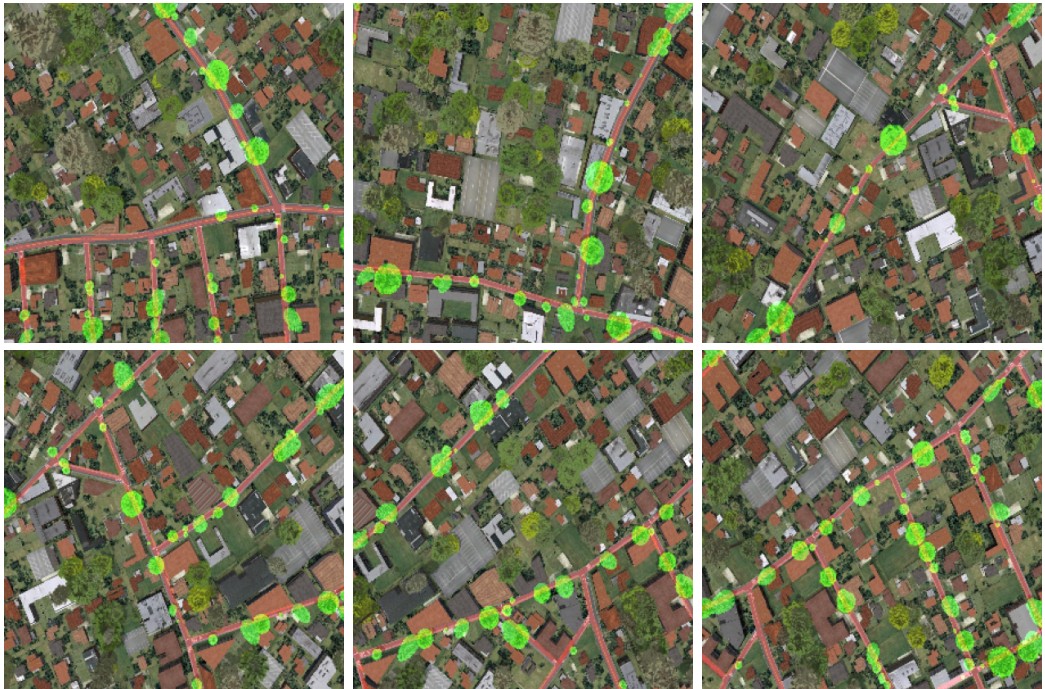

Figure 12: Samples from the synthetic dataset. We generate images and the corresponding street mask, overlaid with the colour red, along with the masks of plants that are occluding the ground truth road network, overlaid with the colour green.

Finally, the architecture of the dynamics network and the value prediction network are shown in Fig. 13. For the value estimation, we also provide the current environment step, as we execute steps in an environment with a bounded time horizon.

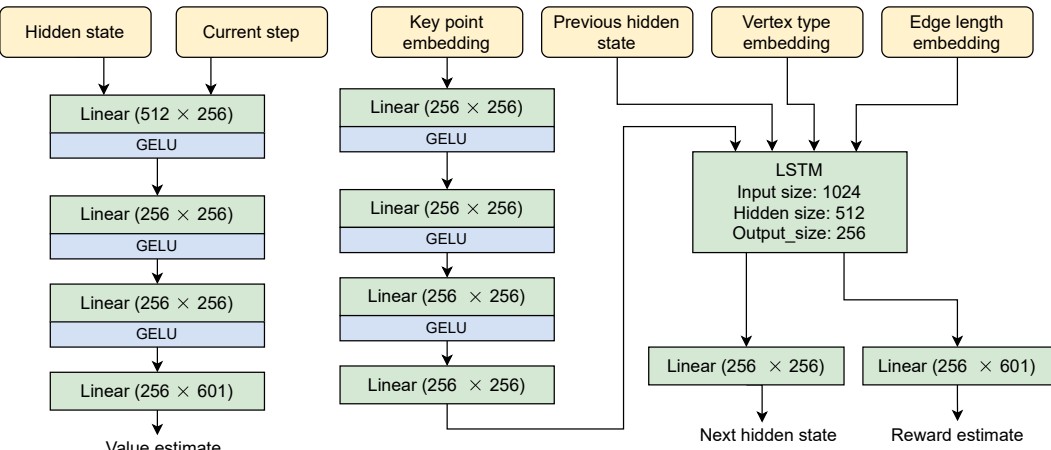

Figure 13: Architecture details of the dynamics network and the value prediction network. Reward and value are determined by predicting support of size 601. Final scalar values are calculated by an invertible transform of this support, similar to Pohlen et al. (2018).

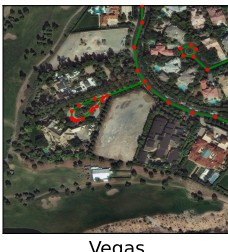 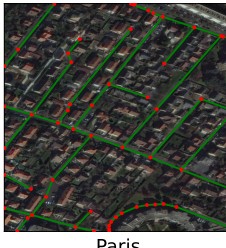 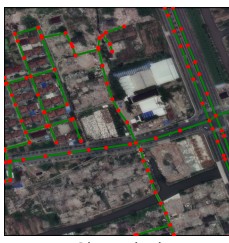 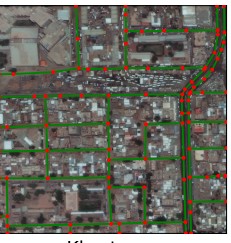

| Vegas | Paris | Shanghai | Khartoum |

Figure 14: Example images sampled from the four cities of the SpaceNet dataset. Images from different cities exhibit different regularity in their road networks. The quality of the overhead satellite images may also vary.

## E  IMPLEMENTATION DETAILS

### E.1  EVALUATION METRICS

*APLS* (Van Etten et al., 2018) constitutes a graph theoretic metric that faithfully describes routing properties. APLS is defined as

$$\text{APLS} = 1 - \frac{1}{N_p} \sum_{p_{v_1 v_2} < \infty} \min \left\{ 1, \frac{|p_{v_1 v_2} - p_{v_1' v_2'}|}{p_{v_1 v_2}} \right\}, \tag{4}$$

where $v$ and $v'$ denote a source node and its closest point on the predicted graph if such exists within a buffer. $N_p$ denotes the number of paths sampled and $p_{v_1 v_2}$ the length of the shortest path between two nodes. Similarly, the *Too Long Too Short (TLTS)* metric (Wegner et al., 2013) compares lengths of the shortest paths between randomly chosen points of the two graphs, classifying them as *infeasible*, *correct*, or too-long or too-short (*2l+2s*) if the length of the path on the predicted graph does not differ by more than a threshold (5%) compared to the ground truth path. Since small perturbations to the predicted graph can have larger implications to pixel-level predictions, the definitions of precision, recall and intersection over union were relaxed in Wiedemann et al. (1998); Wang et al. (2016) leading to the metrics *Correctness/Completeness/Quality (CCQ)*.

Still, some types of errors, such as double roads or over-connections, are not penalized from the above metrics (Citraro et al., 2020). We therefore additionally include new metrics introduced in Citraro et al. (2020) that compare *Paths*, *Junctions* and *Sub-graphs* of the graphs in question, producing respectively *precision*, *recall* and $f_1$ scores. For the final similarity score used in Eq. 3, we use a linear combination of the aforementioned metrics, more details are available in the supplementary material.

### E.2  DATASET INFORMATION

We use the following datasets to train our models, i.e. baselines and our newly proposed RL agent.

**SpaceNet**  (Van Etten et al., 2018) includes a road network of over 8000 Km over four different cities: Vegas, Paris, Shanghai, and Khartoum, where the complexity and quality, and regularity of the road network depend on the city of origin. Satellite images are provided at a pixel resolution of $1300 \times 1300$, corresponding to a ground resolution of $30cm$ per pixel. We split the 2780 total images into crops of size $400 \times 400$ with an overlap of 100 pixels for training. To better highlight the diversity of the satellite images from these four different locations, we have included some randomly sampled examples in Fig. 14.

**DeepGlobe**  (Demir et al., 2018) contains satellite images from 3 different locations with pixel-level annotations. Images have a resolution of $1024 \times 1024$, with a ground resolution of $50cm$ per pixel. We crop the 6226 images into tiles, leading to a similar ground truth resolution per pixel compared to SpaceNet.

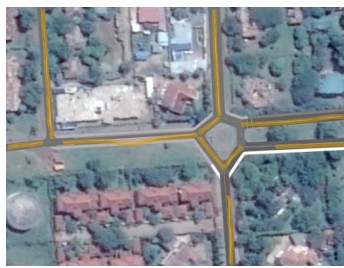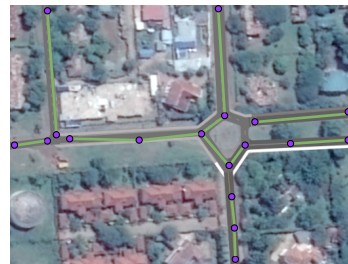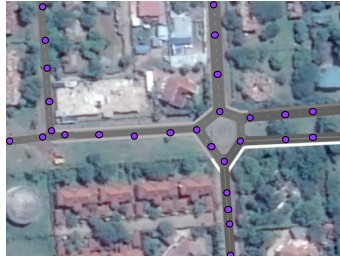

Figure 15: Example of how key points are generated. We start by evaluating a segmentation model (left) and extracting the predicted graph (middle). We then over-sample vertices along edges to enlarge the action space (right).

### E.3 Training details

At each MCTS search step, we perform several simulations from the root state $s^0$ for a number of steps $k = 1, \ldots$ and select an action that maximizes the upper confidence bound (Silver et al., 2018),

$$a^k = \arg\max_a \left[ Q(s,a) + P(s,a) \frac{\sqrt{\sum_b N(s,b)}}{1 + N(s,a)} \left( c_1 + \log \left( \frac{\sum_b N(s,b) + c_2 + 1}{c_2} \right) \right) \right],$$

where $N(s,a), Q(s,a), P(s,a)$ corresponds to the visit counts, mean values and policies, as calculated by the current search statistics. Constants $c_1, c_2$ balance exploration and exploitation. Based on a state $s^{k-1}$ and a selected action $a^k$, a new state $s^k$ and reward $\hat{r}^k$ are estimated through the dynamics network. We update the mean values based on bootstrapped values of the estimated value functions and rewards. We experimented with training the reward and value support predictions with both mean squared error (MSE) and cross-entropy loss. We opted for MSE because of its stability. For a more in depth description of the training scheme of MuZero we recommend Schrittwieser et al. (2020) and Ye et al. (2021).

As hinted in the main text, we train using intermediate rewards, a linear combination of topological metrics. We experimented using a variety of different scores and metrics, but ended up using *APLS*, *Path-based f1*, *Junction-based f1* and *Sub-graph-based f1* at a relative scale of $(0.35, 0.25, 0.25, 0.15)$. We found the *Sub-graph-based f1* to be more sensitive to small perturbations and therefore weighted it less in the final combination. The metrics mentioned above are highly correlated, as examined in Batra et al. (2019). This correlation, though, holds when comparing the final predictions. Intermediate incremental rewards are more independent, so we still found it useful to use a mixture of them. Initially, to let our network learn basic stable rewards, we use the segmentation prediction mask as target. That means that we train our model to predict the graph that can be extracted after post-processing the segmentation model's prediction.

After pre-training the autoregressive model, we experimented with fine-tuning using RL with two different learning rates, where a slower by a factor $(0 - 1]$ rate was chosen for the pre-trained modules. Here, we noticed that the model still performed better than the ARM baseline. As it has trouble though to escape the autoregressive order, compared to the single learning rate model, results are less optimal.

We finally note that by avoiding type and position encoding in the Transformer II module, we can ensure the embedded graph is permutation invariant regarding the sequence of edges and the order of key points within an edge. Our search graph can then be formulated into a directed acyclic graph, circumventing unnecessary division of the search space (Browne et al., 2012; Childs et al., 2008), enabling more efficient sampling (Saffidine et al., 2012). These updated search statistics are cumbersome to compute, though, and we found no significant efficiency improvement. They do, however, confirm our model's potential ability to handle the input graph as an unordered set, as the problem suggests.

### E.4 Producing key points

We initially train a segmentation model for predicting pixel-level accurate masks of the road network. For this step, we can use any model from the literature. We extract the predicted graph by

skeletonizing the predicted mask and simplifying the graph by a smoothing threshold. We then sample intermediate vertices along the largest in terms of ground length edges, to enlarge the action space. We illustrate a toy example of such a process in Fig. 15. To accelerate inference, we can also initialize our prediction graph based on the provided segmentation mask. In such a case, our method closer resembles previous refinement approaches. We additionally remove edges of connected components with small overall size and edges belonging to roads segments leading to dead ends (that means vertices of degree one), keeping though the corresponding key points in the environment state. Thus, if our model deems the existence of the respective edges necessary, it can add them once more. We plan to further investigate augmenting the action space with the ability to remove edges in future work, that would not require such a pre-processing strategy.

### E.5 COMBINING PREDICTIONS

When creating the final per image prediction, we initially simply generated predictions on non-overlapping patches and fused them together. To overcome small pixel location differences in the predicted graphs, we fuse by rasterizing the individual graphs in the pixel domain with a line width larger than 1. What we found more successful was to perform inference on overlapping patches and to initialize the currently predicted graph based on the predictions made so far. This is particularly useful, as road segments are often close to the boundaries of our cropped image. Individual inference and simple fuse can often lead to over-connected predictions. We visualize a toy example of such a process in Fig. 16.

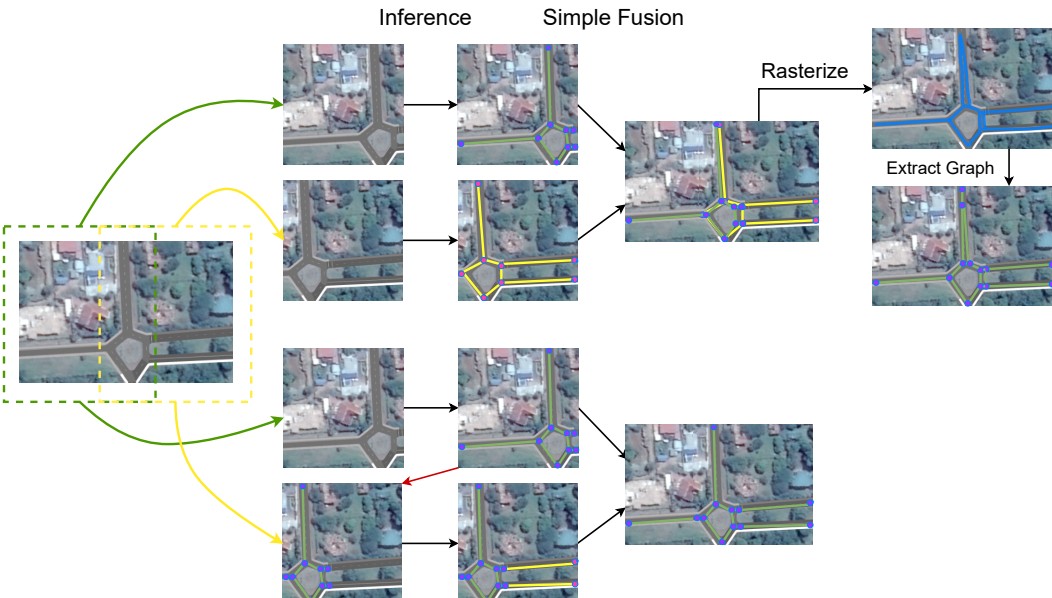

Figure 16: Toy example of how inference is performed in larger images. We start by cropping the image to overlapping patches. (Top) Naive fusion leads to over-connections because of perturbation in the key points' locations. (Bottom) We initialize subsequent graph predictions based on the key points and edge predictions of the so far generated output from previous patches if such exists.

For the segmentation baselines, unless specified in their respective documentation, we perform inference by cropping images to overlapping patches and normalizing the final predicted mask based on the number of overlapping predictions per pixel location. We also pad images around their boundary, as done in Acuna et al. (2019). We note some small differences in the final scores for the Orientation model (Batra et al., 2019) and the SpaceNet dataset, compared to the ones in Citraro et al. (2020). We assume these are an outcome of different chosen parameters for the calculation of metrics. We keep these parameters fixed when calculating scores for all methods.

### E.6 MORE COMPARISONS WITH BASELINES

We elaborate more on the evaluation method on Sat2Graph. The authors provided predictions corresponding only to a center crop of the original SpaceNet dataset images. For each $400 \times 400$ pixel image, predictions are made for the center $352 \times 352$ area of the image. One could expect slightly better results if trained in the same conditions but that the gap does still seem large enough to show the merits of our approach.

Other baselines like Neural turtle graphics (Chu et al., 2019) and Topological Map Extraction (Li et al., 2019b) do not have an implementation available. We do not compare against VecRoad (Tan et al., 2020) or RoadTracer (Bastani et al., 2018), as different datasets were used for the current evaluations. These baselines have been already shown to underperform though in the literature, by methods that we are comparing against.

## F   MORE EXAMPLES

We showcase in Fig. 17 and Fig. 18 more examples of the environment state progression, for the synthetic dataset.

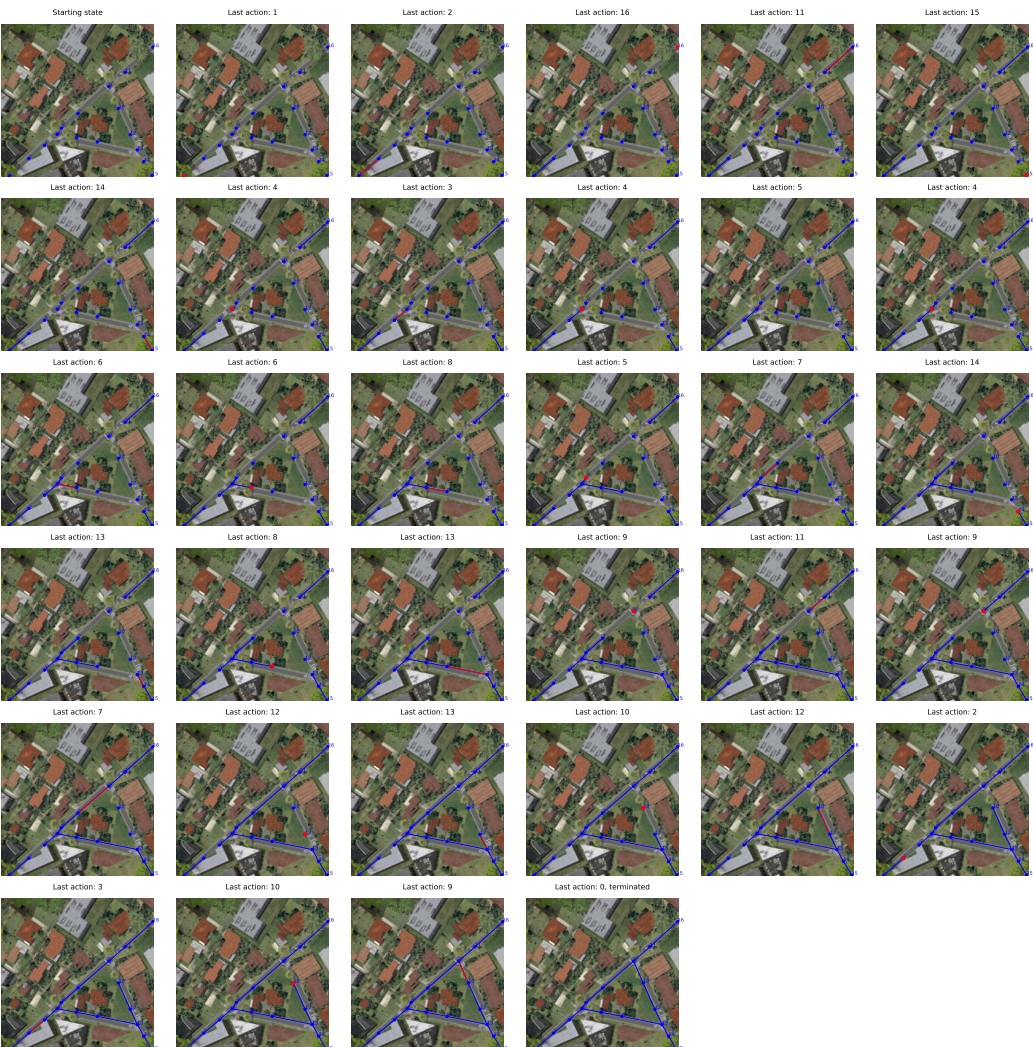

Figure 17: Example of an environment progression for the synthetic dataset. Key points' locations are shown in blue. By over-sampling initial segmentation predictions as shown in Fig. 15, we can generate key points in possibly occluded areas of the image.

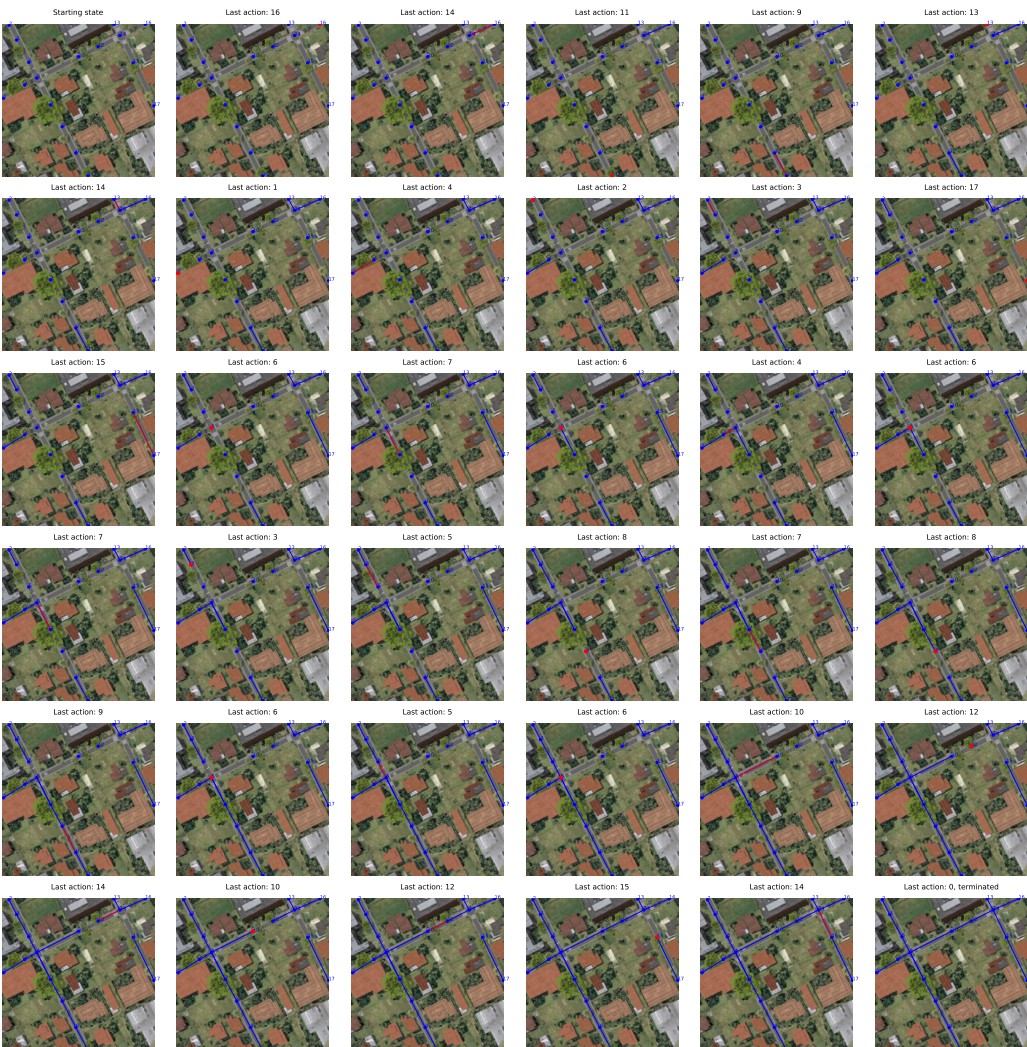

Figure 18: Example of an environment progression for the synthetic dataset. Generating the same edge twice (between key points 6 and 7), although unintuitive, does not lead to a different final predicted graph.

