# OpenReview forum: "Mastering Spatial Graph Prediction of Road Networks"
_ICLR.cc/2023/Conference — Submitted to ICLR 2023_

### Official Review · Reviewer_zSe9 · 2022-10-22

**Confidence:** 3
**Correctness:** 2
**Technical Novelty And Significance:** 3
**Empirical Novelty And Significance:** 2
**Recommendation:** 5

**Clarity, Quality, Novelty And Reproducibility:**

The idea of this paper is original and interesting, however, the authors do not clear show if and how it is generalizable to images with different characteristics. The reviewer was not able to understand the training process (and dataset) used by the authors. It is also unclear if there is any overlap between the training and evaluation datasets.

The presentation style of this work can be improved: in fact, some aspects are described in very generic terms and not in relation to the specific problem (in particular the use of MCTS and the application of MuZero in this situation).

In general, the reproducibility of the work is somehow limited since the description of the use of MuZero and MCTS is only sketched. The steps needed for the training and deployment of the system are also described in very abstract terms.

**Strength And Weaknesses:**

Strengths

- The authors consider an interesting problem with clear practical applications.
- The method used to generate synthetic data is original and interesting.
- It appears that the solution proposed by the authors offers better performance compared to other state-of-the-art methods.

Weaknesses:

- As it is described in the paper, it seems that the proposed method needs the ground truth for actually receiving the reward. It is unclear if the methods is generalizable to different environments. This is not really evaluated in the paper (or at least it is not explicitly described in the paper).
- The reference to MuZero and the description of its application is quite hard to understand, especially in relation to the definition of the rewards. The same applies to the use of a MCTS, which is also only sketched in my opinion.
- The reviewer was not able to understand how the RL is actually trained. In fact, it is unclear if the RL system is evaluated using the same dataset used for training.
- The authors refer a learnable model of a dynamics network G, but it is difficult to map it to the actual inputs (and outputs) of the model.
- The values of the metrics used in the evaluation have very similar values. I believe that the authors should indicate the confidence intervals of these values. The confidence intervals might be overlapping.
- The authors should present the trade-off of the proposed methods in terms of computational complexity. Is the proposed method computationally expensive compared to the existing ones?
- There is limited information about the potential actual deployment post-training in terms of information needed as input. The authors should also clarify the outputs that are provided by the system in a clearer way in my opinion.
- The reviewer was not able to understand the concept of "dreamt tree search" presented in Figure 11.


**Summary Of The Paper:**

The paper describes a solution for learning how to "draw" spatial networks of roads on top of satellite images. The solution is based on MCTS and MuZero. There are limited details in terms of the dataset used for training and how the system is actually deployed after training.

The system is evaluated with synthetic data and real-world satellite images. It is unclear if the proposed solution generalizes to images that are "different" from those used in the training set.


**Summary Of The Review:**

Overall, this is an interesting paper, with potential practical applications. Unfortunately, some key information around the training and deployment of the system are missing. The descriptions of MCTS and MuZero are not completely clear (especially in relation the actions and inputs/outputs of the model).

The performance results do not show significant differences and it is difficult to evaluate them without confidence intervals. Other methods provide better performance in some situations.

It is very difficult to understand if the method generalizes to images that are different from the training set. The authors do not clearly specify if the training and test sets contain similar satellite images.

This is a promising paper, but it is difficult to recommend acceptance in its current version.

---

> ### Author Response · Authors · 2022-11-16
> **Response to Reviewer zSe9**
>
> We thank the reviewer for the precise reading of our work and the comments proposed to improve it. We appreciate that the reviewer acknowledges the practicality of the application and the novelty of the new proposed synthetic dataset.\
> We want to highlight that due to space limitation issues, we have deferred more details regarding datasets, training, and inference in the appendix. We also welcome the reviewer to examine the code that we have released as part of the supplementary material for further details. The code will also be made publicly available.
>
> 1. **Synthetic dataset:** We thank the reviewer for raising issues of generalization of our methodology. This is especially important given the application, as we cannot expect to always have enough training data for each new area our method would be applied to. We argue that: (1) The SpaceNet [1] dataset used already exhibits a lot of diversity, as images are collected from 4 different cities with different characteristics and complexities. We have better highlighted the **diversity of these satellite images** in the appendix by also including some randomly sampled images for each city in Fig. 14. (2) In the paper we already show that **our method generalizes in unseen datasets**. More specifically, for the DeepGlobe dataset, we do not fine-tune our model but instead, just directly perform inference in this new dataset. We only need to standardize images differently according to the new per-channel statistics. As you can see by the results in Table 1, our model is able to generalize adequately even without specific fine-tuning. This is due to the fact that in general, road network statistics and characteristics (e.g. nature of intersections and their density), can be expected to be somewhat consistent across datasets.
> An additional factor is that Transformers [2], on which our basic autoregressive model is based upon, are able to generalize better in relationship modelling tasks (e.g. [3]), which is how we have framed the problem in this work. This is because Transformers generalize by information stored in the weights, but also information stored in the context itself [4] (i.e. currently generated edge list).
> 2. **Required information:** We use the ground truth label information **only** to compare the currently proposed road-network graph with the true one, through Eq. (3). More specifically, if a modification to the predicted graph leads to higher similarity to the ground-truth graph then the reward is positive, otherwise negative. The similarity function can be **arbitrarily complex**, and the network is agnostic to its specificity. In this sense, our method is general enough to be applied to any environment where such intermediate rewards can be generated. This corresponds to the standard reinforcement learning regime.
> 3. **Description of MuZero:** We thank the reviewer for requesting more details regarding MuZero. At the heart of MuZero is an interventional world model that predicts the one-step dynamics of the world and its intermediate rewards. MuZero produces forward simulations of the world, producing imagined state-reward sequences. MuZero also models a value function and a probabilistic policy by a shared DNN. The details of all these networks are given in Appendix D. Reward-related prediction errors are then used to train these networks. Here, we define as intermediate rewards the improvements of the currently generated graph resulting from the last action, through Eq. (3). A detailed description of all the rewards used in this work is provided in Appendix E1. During training, MCTS is used to explore actions based on a tradeoff between exploitation and exploration. We have included some of these details in Appendix E3. For more information, we direct the reviewer to the excellent description of [5] or [6] in Appendix A3.
> 4. **Training and evaluation of RL system:** We point the reviewer to Appendix E2 (due to space limitations) for details regarding the datasets used, which we have rewritten to make more clear. More specifically, the RL system is trained on a train split of the SpaceNet dataset and evaluated on a held-out validations set. As mentioned in the main text, we use the same train-test splits as in [7] to make comparisons easier. We also evaluate our agent on the DeepGlobe dataset (again using the splits from [7]), without any additional fine-tuning.
> 5. **Dynamics network:** The dynamics network produces two outputs. One-step forward dynamics of the world in the model’s latent space and intermediate predicted rewards. This hidden state of the model is the one visible in the right part of Figure 4, describing the edges embedding.

---

> > ### Author Response · Authors · 2022-11-16
> > **Response to reviewer zSe9 (cont)**
> >
> > 6. **Evaluation scores:** We thank the reviewer for pointing out that some evaluation scores are similar to existing approaches. We highlight however that in what **graph-theoretic metrics** are concerned, our approach consistently **outperforms** all baselines. This is also prevalent in the generated graph statistics of the different methods in Figure 7 and by considering road networks with increasing difficulty in Figure 10 of the appendix. Unfortunately, computing confidence intervals for all methods is computationally too expensive. It is generally common in these applications to not report confidence intervals (e.g. [7, 10]). We hope that the large size of these datasets (both within the train and the test split) compensates for potential expected noise in the results.
> > 7. **Computational complexity trade-off:** We agree with the reviewer that the computational complexity trade-off is a very important one. This same trade-off motivated us to introduce ablation studies, where we consider a smaller number of simulations in Figure 8 (right). Fewer simulations lead to faster training and inference. During training, we use the same number of GPUs with existing methods [7]. We acknowledge however that our method requires also CPU agents to collect trajectories based on the current policy to eventually train our agent.
> > 8. **Deployment:** The whole pipeline does not require more information than the satellite image that the road network should be generated for. It then predicts a spatial graph where nodes denote specific locations in the image and edges correspond to roads predicted in the image. Note that this corresponds to the mapping standards of map repositories (e.g. OpenStreetMap [8], ArcGIS StreetMap [9]) and labelling standards of road network datasets (e.g. SpaceNet). Segmentation approaches need an additional post-processing step to get the predicted output to the desired format. We have included examples of this generation process for the toy dataset in Fig. 17 and 18 in the appendix.
> > 9. **Dreamt tree search in Figure 11:** As previously mentioned, MuZero is an interventional world model that predicts the one-step dynamics of the world. This means that it simulates sequences of actions (in the form of a tree) in a “dreamt” latent space. In Figure 11, for illustration purposes, we decode these sequences of actions to the corresponding environment state. Again, as stated in the figure caption, the model does not have access to these environment states. The depicted figure is the tree search generated before performing a single action from the current state (the root of the tree).
> >
> > From our reading, the reviewer’s initial concerns were mostly about clarity. Given that we have better described the methods used and updated the text to reflect the reviewer’s suggestions, we would like to ask that the reviewer reconsiders their final evaluation. We are more than happy to engage in further discussion regarding further considerations.
> >
> >
> > [1] Adam Van Etten, Dave Lindenbaum, and Todd M Bacastow. Spacenet: A remote sensing dataset and challenge series. arXiv preprint arXiv:1807.01232, 2018.
> >
> > [2] Vaswani, Ashish, et al. "Attention is all you need." Advances in neural information processing systems 30 (2017).
> >
> > [3] Kim, Jinwoo, Saeyoon Oh, and Seunghoon Hong. "Transformers Generalize DeepSets and Can be Extended to Graphs & Hypergraphs." Advances in Neural Information Processing Systems 34 (2021): 28016-28028.
> >
> > [4] Chan, Stephanie CY, et al. "Transformers generalize differently from information stored in context vs in weights." arXiv preprint arXiv:2210.05675 (2022).
> >
> > [5] Schrittwieser, Julian, et al. "Mastering atari, go, chess and shogi by planning with a learned model." Nature 588.7839 (2020): 604-609.
> >
> > [6] Ye, Weirui, et al. "Mastering atari games with limited data." Advances in Neural Information Processing Systems 34 (2021): 25476-25488.
> >
> > [7] Batra, Anil, et al. "Improved road connectivity by joint learning of orientation and segmentation." Proceedings of the IEEE/CVF Conference on Computer Vision and Pattern Recognition. 2019.
> >
> > [8] OpenStreetMap contributors. Planet dump retrieved from https://planet.osm.org . https://www. openstreetmap.org, 2017.
> >
> > [9] Sources: Esri, DigitalGlobe, GeoEye, i-cubed, USDA FSA, USGS, AEX, Getmapping, Aerogrid, IGN, IGP, swisstopo, and the GIS User Community
> >
> > [10] Xu, Zhenhua, et al. "Rngdet: Road network graph detection by transformer in aerial images." IEEE Transactions on Geoscience and Remote Sensing 60 (2022): 1-12.

---

> > ### Comment · Reviewer_zSe9 · 2022-12-04
> > **Thanks for the responses**
> >
> > Thanks for the responses - please find some notes below.
> >
> > 1. thanks for clarifying this - but in my opinion, it is difficult to understand how these experiments demonstrate generalizability (limited informations provided about the detasetes).
> > 2. I wonder if the comparison is not valid since the available information is different?
> > 3. Thanks for adding this information, but the actual theoretical foundations of the model is still not completely clear (especially in terms of forward simulations of the world).
> > 4. Thanks for adding this information.
> > 5. Thanks for adding these details - quite helpful.

---

> > > ### Author Response · Authors · 2022-12-08
> > > **Response to Reviewer's Feedback**
> > >
> > > We thank the reviewer for the feedback and the constructive conversation.
> > >
> > > 1. (a) The additional benefit of the synthetic dataset is that we have exact information regarding the location of the occlusions. This is in sharp contrast to existing road network datasets where such knowledge is unattainable (one can try to predict the existence of vegetation or other buildings occluding the road network, but this will be inexact). By utilizing the synthetic dataset, we can characterize the type of mistakes made in the predictions, i.e. the severity of the occlusions at the location of mistakes. We note that details regarding the synthetic dataset are provided in Appendix C. Exact code/instructions to reproduce the synthetic dataset are also provided in the supplementary material, under the 'CityEngine' folder. (b) For the results on the DeepGlobe dataset displayed in Table 1, we do not finetune our model on the DeepGlobe dataset, but instead just perform inference based on the pretrained (on the SpaceNet dataset) model. This showcases that our model successfully captures road statistics that are present across datasets and is able to transfer this knowledge to new unseen datasets. Our model relies less on pixels values but instead on the graph properties of the road network. The SpaceNet and DeepGlobe datasets are widely used in the literature, we have included more information regarding those in Appendix E2, due to space limitation.
> > > 2. Training requires that one provides the ground truth information. Ground truth information in the form of a spatial graph is usually available in map repositories, also the SpaceNet dataset. Even in the case that such information is not explicitly provided, but instead we are given the ground truth segmentation mask, we can still extract an approximate ground truth spatial graph, via a skeletonizing process. This is what is actually being done for the DeepGlobe dataset, as only the segmentation ground truth masks are provided. This process can be portrayed in the folder ‘rl_road_extraction/dataprocessing/’, which is heavily inspired by the code of Batra et al. (acknowledged in the README). Our method thus **does not require** any additional information. Finally, note that previous segmentation approaches (e.g. Batra et al.) require segmentation ground truth masks. When this information is not provided (e.g. for the SpaceNet dataset), these methods require the rasterization of the spatial graph into a segmentation mask, with a constant width which is an additional hyper-parameter.
> > > 3. Forward simulations in MuZero (via the dynamics network) estimate new hidden states and rewards. The main motivation is that this world model can be used instead of running expensive steps in the real environment, and so we bypass the expensive call to the decoder module during the search, and can instead approximate small modifications in the latent representation directly. The dynamics model is small in size, and as manifested in our paper, it requires up to 90 times less floating-point operations to simulate trajectories, compared to using the edge embeddings’ decoder, accelerating training significantly.

---

### Official Review · Reviewer_BpnF · 2022-10-24

**Confidence:** 4
**Correctness:** 4
**Technical Novelty And Significance:** 4
**Empirical Novelty And Significance:** 4
**Recommendation:** 8

**Clarity, Quality, Novelty And Reproducibility:**

The approach shows as novel and the results are good. I view high novelty.

The code and data are referred to in the text. The work should be reproducible.

**Strength And Weaknesses:**

The paper is well-written and easy to follow.
The literature is covered well.
The method is very useful for road network extraction and presents a solution to a real-world problem.
The results are strong with additional results provided in the appendix.

The placement of figures is out of place and I don't see all figures referred to in the text. The text doesn't often tell the story alongside the figures.

In remote areas, the target of this work, roads are often informal and exhibit a lot of variability e.g. width, surface, and use. Has this been considered?  A centerline wouldn't work in such cases.


**Summary Of The Paper:**

The paper presents a road network prediction method using spatial graphs. The approach uses reinforcement learning.

A number of graph measures are used to assess the method.

**Summary Of The Review:**

I thoroughly enjoyed this paper - the objective is clear, and the methodology and results well done.

---

> ### Author Response · Authors · 2022-11-16
> **Response to Reviewer BpnF**
>
> We thank the reviewer for the precise and thorough reading of our work. We are very glad to hear that our work provides novel and interesting insights into the task of road network extraction, an important task with many implications in both urban and rural communities. A lot of work and care was also devoted to providing detailed descriptions of all techniques and choices made by an extensive appendix.
> 1. **Placement of figures:** Thank you for pointing out ways to improve the quality of the text and the storytelling. We have rearranged figures and parts of the text to better highlight important points in our story and help the reader understand intuitively our methodology. All figures in the main text and supplementary material are now explicitly referred to.
> 2. **Variability in roads:** We thank the reviewer for pointing out the possible variability of road networks. The final prediction of our network is a graph, with nodes indicating intersections and edges denoting connections between them. Note that this corresponds to the mapping standards of map repositories (e.g. OpenStreetMap [1], ArcGIS StreetMap [2]) and labelling standards of road network datasets (e.g. SpaceNet [3]). We highlight that part of the information regarding the type of each road is implicitly included in the dataset our model is trained on. For example, wide roads (highways) are encoded as multiple parallel roads in the spatial graph (Figure 3-right on the revised paper PDF). We argue that **our model captures such knowledge** encoded in the dataset, a fact also highlighted by comparing the generated road network statistics in Figure 7. Additionally, the fact that we are directly predicting road networks as graphs, makes it easy to predict meta-information (additional features) for the extracted nodes and edges, including, for instance, the type of road (highway, primary or secondary street, biking lane, etc). Given datasets that include this type of information, we claim that our method could be trivially enhanced to produce such predictions in the form of additional edge features. These could correspond for instance to road width or road type. Google maps include such types of features (https://support.google.com/mapcontentpartners/answer/160414?hl=en).
>
> We hope that we have answered the reviewer’s concerns and are more than happy to engage in further discussion regarding considerations that the reviewer may have.
>
> [1] OpenStreetMap contributors. Planet dump retrieved from https://planet.osm.org . https://www. openstreetmap.org, 2017.
>
> [2] Sources: Esri, DigitalGlobe, GeoEye, i-cubed, USDA FSA, USGS, AEX, Getmapping, Aerogrid, IGN, IGP, swisstopo, and the GIS User Community
>
> [3] Adam Van Etten, Dave Lindenbaum, and Todd M Bacastow. Spacenet: A remote sensing dataset and challenge series. arXiv preprint arXiv:1807.01232, 2018.

---

### Official Review · Reviewer_ro23 · 2022-10-25

**Confidence:** 3
**Correctness:** 4
**Technical Novelty And Significance:** 2
**Empirical Novelty And Significance:** 2
**Recommendation:** 6

**Clarity, Quality, Novelty And Reproducibility:**

The quality of the presentation part is good and the writing is clear.
The quality from the methodological perspective seems to be good too.
The idea seems to be novel.
Supplementary materials with the code enable the reproducibility of experiments.


**Strength And Weaknesses:**

Strengths:
- the introduced methodology seems to be quite novel
- the description of the methodology and experiments is comprehensive and the writing is quite good and clear
- supplementary materials with the code which enable the reproducibility of experiments

Weaknesses:
- I have doubts regarding the importance of the use case (road network modelling from images) because nowadays, digital descriptions of road networks are usually generally available, but there are other works dealing with this topic and perhaps it's a good use case to demonstrate the introduced methodology. However, it would be good to demonstrate that the method can be useful in some other applications too.

**Summary Of The Paper:**

The paper introduces a reinforcement learning framework for generating a graph as a variable-length edge sequence, where a structured-aware decoder selects new edges by simulating action sequences into the future. It reviews related works, explains the methodology, and presents the settings and results of experiments. The paper is followed by the Appendix with more details about the architecture, implementation, dataset creation and interoperability. There are also supplementary materials with the code which enable the reproducibility of experiments.

**Summary Of The Review:**

The paper introduces a reinforcement learning framework for generating a graph as a variable-length edge sequence, where a structured-aware decoder selects new edges by simulating action sequences into the future. It reviews related works, explains the methodology, and presents the settings and results of experiments. The paper is followed by the Appendix with more details about the architecture, implementation, dataset creation and interoperability. The idea seems to be novel. The quality of the presentation part is good and the writing is clear. The quality from the methodological perspective seems to be good too. There are also supplementary materials with the code which enable the reproducibility of experiments. I have doubts regarding the importance of the use case (road network modelling from images) because nowadays, digital descriptions of road networks are usually generally available, but there are other works dealing with this topic and perhaps it's a good use case to demonstrate the introduced methodology. In the future, it would be good to demonstrate that the method can be useful in some other applications too. All in all, I like the paper and in my opinion, it's good enough to be published at ICLR.

---

> ### Author Response · Authors · 2022-11-16
> **Response to Reviewer ro23**
>
> We thank the reviewer for the precise and thorough reading of our work. We are happy to hear that our method provides some novel insights into methodologies to construct spatial graphs. Furthermore, we are glad to see that the extensive methodology description and the supplementary material help with having a clear understanding of the proposed methodology and specific methods used for the different components of our work. A lot of effort was devoted to making the supplementary material as concrete and detailed as possible.
>
> We thank the reviewer for pointing out potential future work and future applications for the newly proposed methodology. As indicated by the reviewer, the methodology is general and can be applied to any application that requires the **generation of a (spatial) graph**. We give two such examples/applications where our methodology can be applied out of the box.
> - Scene Graph Generation (e.g [1, 2]): The task here is to identify the contextual information between objects and the relationships between them. Similar to our setting, an object detection model proposes possible objects recognized in the scene and propagates visual features corresponding to each object separately. Our framework then identifies relationships between the objects by decoding the sequence of edges between selected objects, exhibiting the same advantages in this case compared to the autoregressive decoding solution. Note that our method can be applied both to the 2D, as well as the 3D regime. Observe also that our method can scale to many points (nodes in the graph) by also using hierarchical representations (e.g [3]).
> - Visual reasoning/Factual Visual Question Answering (e.g. [4, 5]). Similar to the previous case, we want to accurately encode relationships (between a potentially large set of possible values) between objects/actors.
>
> We argue, however, that our method is particularly well-suited for the task of road network extraction. More specifically, accurate predictions require the ability to **interactively focus on different parts of the image**. We are also able to **directly optimize non-continuous rewards** which correspond exactly to the graph metrics used to evaluate the final predicted road network. This constitutes a novel approach for this application. Previous works have mentioned fundamental difficulties that make road network extraction a particularly challenging task (e.g. [6]). We are able to cope with these challenges due to the flexibility of our method, which is reflected by the results on graph metrics in Table 1 and the extracted statistics in Figure 7. Previous attempts to deal with such problems, as well as the problem of producing discontinuous predictions in general, were mainly focused on applying specific post-processing, something that our method does not need, as it produces outputs directly in the required format. \
> Finally, although digital descriptions of road networks are generally available from big map repositories (OpenStreetMap [7], Google Maps), this does not always hold for developing countries [8] or for roads exhibiting frequent changes. Changes in the road network can be planned, e.g. expansion of the road surface, or unplanned, for instance in the form of a natural catastrophe, e.g. flood [9]. In any case, we would like for the road network to be constantly up-to-date, among others for navigation purposes and to accommodate self-driving vehicles.
>
> We hope that we have answered the reviewer’s concerns and are more than happy to engage in further discussion regarding further considerations that they may have.

---

> > ### Author Response · Authors · 2022-11-16
> > **Response to Reviewer ro23 (cont)**
> >
> > [1] Xu, Danfei, et al. "Scene graph generation by iterative message passing." Proceedings of the IEEE conference on computer vision and pattern recognition. 2017.
> >
> > [2] Yang, Jianwei, et al. "Graph r-cnn for scene graph generation." Proceedings of the European conference on computer vision (ECCV). 2018.
> >
> > [3] Landrieu, Loic, and Martin Simonovsky. "Large-scale point cloud semantic segmentation with superpoint graphs." Proceedings of the IEEE conference on computer vision and pattern recognition. 2018.
> >
> > [4] Chen, Xinlei, et al. "Iterative visual reasoning beyond convolutions." Proceedings of the IEEE conference on computer vision and pattern recognition. 2018.
> >
> > [5] Narasimhan, Medhini, Svetlana Lazebnik, and Alexander Schwing. "Out of the box: Reasoning with graph convolution nets for factual visual question answering." Advances in neural information processing systems 31 (2018).
> >
> > [6] Batra, Anil, et al. "Improved road connectivity by joint learning of orientation and segmentation." Proceedings of the IEEE/CVF Conference on Computer Vision and Pattern Recognition. 2019.
> >
> > [7] OpenStreetMap contributors. Planet dump retrieved from https://planet.osm.org . https://www. openstreetmap.org, 2017.
> >
> > [8] Silyanov, V. V., et al. "An overview road data collection, visualization, and analysis from the perspective of developing countries." IOP Conference Series: Materials Science and Engineering. Vol. 832. No. 1. IOP Publishing, 2020.
> >
> > [9] Rahnemoonfar, Maryam, et al. "Floodnet: A high resolution aerial imagery dataset for post flood scene understanding." IEEE Access 9 (2021): 89644-89654.

---

### Official Review · Reviewer_d6d1 · 2022-10-25

**Confidence:** 4
**Correctness:** 3
**Technical Novelty And Significance:** 2
**Empirical Novelty And Significance:** 2
**Recommendation:** 3

**Clarity, Quality, Novelty And Reproducibility:**

Clarity/quality:

As previously stated, the pipeline consist of a number of methods mashed together. Even an overview is missing - Fig 1 describes muZero modeling, which is a tiny bit of the whole pipeline. IMHO the authors need to test a modern architecture, such as RNGDet and check if RL improves on top of that. If not, well, RL is not the way to go for this task. Especially looking at the RNGNet results, I do not see compelling evidence the authors would be able to come close to those numbers, even with the complex system they built on top of the (already existing) graphs.

The paper has only minor text issues - e.g., Fig 6. "imporovement" x2, it is well written and structured.

Novelty:

All in all, I do not think this paper in its current form is ICLR-worthy. It is more of an engineering work that struggles to marginally improve the results. Judging from the ablation study, they start from a terrible baseline, which again, in 2022, it is unacceptable. If my calculator is right, the 0.455 ALPS is the method without any improvements. Why? If the focus is the RL agent, why not start with the best graphs possible and improve over them? Is it something here that I am missing?

The synthetic dataset could be used to improve the results and show a compelling advantage for RL, but it's not. In fact, it's not even clear on how it's used, they claim the generation of a 'difficulty' score and training with LinkNet [a 2017 method], but only Figure 6 supports this claim, and no other attempts to find a meaningful use are described (e.g., is the autoregressive pretrained done on this dataset? What about other ideas, such as style transfer, or use to train the RL agent? Or is it this already done but not mentioned?).

Few recent approaches have attempted to upgrade the algorithms to modern pipelines (e.g., directly output vertices similar to [1*,2*], without intermediate keypoints/road segmentation, as in RNGDet/RNGDet++[3*- this paper is here only for future reference]), it would have been nice to adapt the framework, but no efforts were made.



Reproducibility:

The code is released in the supplementary material, it should be easy to reproduce the experiments.

___
[1*]Lazarow, J., Xu, W., & Tu, Z. (2022). Instance Segmentation With Mask-Supervised Polygonal Boundary Transformers. In Proceedings of the IEEE/CVF Conference on Computer Vision and Pattern Recognition (pp. 4382-4391).

[2*]Liang, J., Homayounfar, N., Ma, W. C., Xiong, Y., Hu, R., & Urtasun, R. (2020). Polytransform: Deep polygon transformer for instance segmentation. In Proceedings of the IEEE/CVF Conference on Computer Vision and Pattern Recognition (pp. 9131-9140).

[3*]Xu, Z., Liu, Y., Sun, Y., Liu, M., & Wang, L. (2022). RNGDet++: Road Network Graph Detection by Transformer with Instance Segmentation and Multi-scale Features Enhancement. arXiv preprint arXiv:2209.10150.

**Strength And Weaknesses:**

### Strengths
- competitive performance on two public datasets
- interesting synthetic dataset with path occlusion difficulty analysis

### Weaknesses
- more recent [relevant] related work that yields much improved results such as RNGDet (ignored) or VecRoad (different split, mentioned but ignored) is not compared against; what do you mean by "these baselines have been already shown to underperform though in the literature"? Who has shown that and where?
- small performance improvements on both datasets, seems to struggle at times, sometimes beaten by a 2017 paper
- convoluted pipeline, RGB segmentation >> point extraction from sampling >> auto-regression model >> muZero; feels like the whole system was glued in place to fix the problems from the previous step (e.g., the sampled keypoints are not perfect, throw in the auto-regression transformer model; its graphs are not that great, throw in the RL to figure out which graphs are usable); why not try predicting the graph directly?
- using RL for navigation / computing routes is not novel; for example, [1*] presented a similar concept, but with street views instead of satellite images.


___
[1*]Mirowski, P., Grimes, M., Malinowski, M., Hermann, K. M., Anderson, K., Teplyashin, D., ... & Hadsell, R. (2018). Learning to navigate in cities without a map. Advances in neural information processing systems, 31.

**Summary Of The Paper:**

The authors propose a reinforcement learning approach to road vectorization. In contrast with previous works, the task is modeled as generating a graph as a variable-length edge sequence.

The pipeline consists of multiple stages: semantic segmentation from RGB images (this results in binary maps), a transformer-based autoregression model (results in a collection of road graphs) and the RL part- muZero adapted for road graph extraction. It achieves competitive performance on DeepGlobe and SpaceNet datasets.

The authors also introduce a synthetic dataset that helps pretraining the autoregression model (need to clarify with the authors on that).

**Summary Of The Review:**

The paper provides an interesting alternative to all-in-one transformer-based architectures, but the latter are not compared and show significant improvements compared to (Batra et al., 2019), even though the split is different - RNGDet ( Xu et al., 2022)] or at least VectorNet (Tan et al., 2020). The ablation study is worrying to say the least - for an unknown reason, they start with a very poor baseline, despite the focus of the paper being the novel RL graph fixing method. I am waiting for the authors to clarify on this, I hope I haven't understood something right.

That being said, it proposes an interesting synthetic datasets approach for modelling occlusions/ambiguity. Nevertheless, it is underexploited - not clear why an old road detection was trained on it and how it helps the RL. It was only used in Figure 6 to check that LinkNet (a 5 year old method) is worse?

---

> ### Author Response · Authors · 2022-11-16
> **Response to Reviewer d6d1**
>
> We thank the reviewer for the precise and thorough study of our work. We are happy to hear that our method provides novel insights into the road-network extraction task.
>
> 1. **Synthetic dataset:** The synthetic dataset is not used as a pretraining step for our method. We have updated the text to make this more clear. The purpose of the synthetic dataset is to highlight the point that segmentation models can struggle in the case of severe occlusion, whereas our method does not. We use a synthetic dataset to generate masks for the occlusions themselves, so we can classify the nature of the mistakes and the difficulty of each synthetic satellite image, based on the percentage of occlusion. Figure 6 illustrates the comparisons within this synthetic dataset. The rest of the experiments reflect training and evaluation on real datasets (SpaceNet and DeepGlobe).\
> We motivate our choice of the baseline (LinkNet) as the most successful segmentation approach for the task of road network extraction. We by no means argue that this baseline constitutes a state-of-the-art method. We merely use it as a representative of the models within the “segmentation class”. Note finally, that LinkNet, with small variations and enhancements, has been actively used in many recent approaches, e.g. [1-4].
> 2. **Additional related work:** We thank the reviewer for pointing out more recent related work, which we have now included in the related work section. In general, we have compared with all relevant baselines that provided source code. For papers without code, it is of course impossible to establish a reliable comparison. We note that RNGDet as of the 14th of November does not have an available implementation for training (only for inference for the RoadTracer dataset). We on the other hand will release our code.
> RNGDet [5], which appeared on arxiv earlier this year, is trained using imitation learning, meaning that they provide direct supervision to the model’s prediction. This is in sharp contrast to our scenario where we are only providing non-differentiable, potentially sparse rewards to our agent for selected actions. We are thus able to **directly optimize based on the metrics of interest**. Furthermore, **we do not specify an order of traversing the nodes** of the road network graph, but instead, allow the agent to attend to any parts of the image that they consider easier to start with. Finally, RoadTracer has been shown to underperform against Sat2Graph for the SpaceNet dataset that we are also comparing against in [6].
> 3. **Results:** We compare against a range of successfully adopted methods in the literature. The 2017 paper only beats us once, and this is due to the trade-off between recall and precision for some of these metrics. We also remark that it is common for a method to be beaten in some evaluation scores, e.g. RNGDet (TABLE I) seems to severely underperform for some of the reported metrics. We highlight that in graph-theoretic metrics, our method consistently outperforms recent baselines. This also holds for the case that we do not specifically fine-tune on a specific dataset (DeepGlobe). On the contrary, we train our baselines from scratch on DeepGlobe. Showing that our method exhibits generalization across datasets is especially important, as we cannot expect to have enough data for each new area of interest separately.
> 4. **Pipeline:** We do have several steps in our pipeline, but these are automated and run independent of user interference. We note that our method is robust and generalizes to different datasets, even without additional fine-tuning. It also produces outputs as **graphs directly**.
> 5. **Connections to existing literature:** We thank the reviewer for pointing out the work in [7]. Although the idea of computing routes is similar, the fundamental ideas and the application are completely different. First, [7] uses a fixed number of actions that correspond to specific movements of the agent. Our **action space is of variable size** and depends on the number of key points provided. Then, they greedily interact with the city environment to go from an initial starting position to the final goal. Although this might make sense for navigation planning, extracting a road networks can benefit from choosing the location of the image to act upon first, which we do. Finally, we highlight once again that the graph-theoretic metrics we are using are challenging and may be highly discontinuous and sparse. \
> One observation they make in [7] is that the agent benefits from using a training curriculum to iteratively make goals harder and harder. As defining easier goals for our task is more difficult, we instead embark on autoregressive pre-training to provide some initial useful signal to the agent. Eventually, we significantly outperform this autoregressive baseline though, see Appendix A and Table 3.
> 6. **Typo:** Thank you for pointing out the typo in Figure 6, now fixed.

---

> > ### Author Response · Authors · 2022-11-16
> > **Response to Reviewer d6d1 (cont)**
> >
> > 7. **Ablation study:** In the ablation study, we study the importance of different components of our method and their implications for the graph-theoretic metrics we are using to evaluate. In short, the main findings are the following; (1) Using autoregressive decoding instead of the RL framework is suboptimal. In practice, we found that the autoregressive model often chose to terminate sequences prematurely. As it is trained in a supervised manner, when put in a state that the agent has not seen before, the behavior is undefined. RL improves on that via better exploration of the environment. (2) Including visual features in the key points embeddings is beneficial for the model. (3) Tree-search also helps to improve the initial policy even during inference. This is in line with results reported in the literature, e.g. [8]. (4) Including additional image features does not seem to lead to better performance in our case.\
> > The baseline in this case does not refer to starting with any particular graph predictions. It instead refers to removing some components of our proposed methodology.
> >
> > We hope that we have removed any confusion regarding the specifics of the **methodology** and the **novelty** of our approach. More specifically, we clarified specifics on (1) the components of our method, (2) the synthetic dataset, and (3) connections with additional related work. We have updated the text to reflect these changes and added related work proposed by the reviewer. Given these new insights, we would like to ask the reviewer to reconsider their final evaluation. We are more than happy to engage in further discussion regarding further considerations.
> >
> > [1] Li, Ruirui, Bochuan Gao, and Qizhi Xu. "Gated auxiliary edge detection task for road extraction with weight-balanced loss." IEEE Geoscience and Remote Sensing Letters 18.5 (2020): 786-790.
> >
> > [2] Wang, Yooseung, Junghoon Seo, and Taegyun Jeon. "NL-LinkNet: Toward lighter but more accurate road extraction with nonlocal operations." IEEE Geoscience and Remote Sensing Letters 19 (2021): 1-5.
> >
> > [3] Zhang, Ju, et al. "Learning from GPS trajectories of floating car for CNN-based urban road extraction with high-resolution satellite imagery." IEEE Transactions on Geoscience and Remote Sensing 59.3 (2020): 1836-1847.
> >
> > [4] Chen, Ziyi, et al. "Road extraction in remote sensing data: A survey." International Journal of Applied Earth Observation and Geoinformation 112 (2022): 102833.
> >
> > [5] Xu, Zhenhua, et al. "Rngdet: Road network graph detection by transformer in aerial images." IEEE Transactions on Geoscience and Remote Sensing 60 (2022): 1-12.
> >
> > [6] He, Songtao, et al. "Sat2graph: Road graph extraction through graph-tensor encoding." European Conference on Computer Vision. Springer, Cham, 2020.
> >
> > [7] Mirowski, P., Grimes, M., Malinowski, M., Hermann, K. M., Anderson, K., Teplyashin, D., ... & Hadsell, R. (2018). Learning to navigate in cities without a map. Advances in neural information processing systems, 31.
> >
> > [8] Schrittwieser, Julian, et al. "Mastering atari, go, chess and shogi by planning with a learned model." Nature 588.7839 (2020): 604-609.

---

### Author Response · Authors · 2022-11-16
**To All Reviewers**

We thank all the reviewers for the careful study of our work and the constructive feedback. In the following, we will address the questions shared by all the reviewers.

**Clarity:** In the updated paper PDF we have reformulated certain parts of the paper and rearranged figures so that they closely follow the flow of the text and the storyline.

**Novelty:** Our proposed method is trained on a reward function that captures complex, non-continuous metrics that are directly connected to the application in question. It attends globally to the whole image and does not require a specific order when generating predictions. In addition, it can be used to refine predictions from another given model. The final outputs are in the required graph format directly, enabling possibly the concurrent prediction of meta-information about the edges, e.g. type of the road.

**Other applications:** We have presented our approach for the task of road-network extraction. In reality, the methodology we are proposing can be generalized and easily applied to other applications. We refer to our answer to reviewer ro23 for more details regarding such possible applications and our motivations for the specific application, road network extraction from satellite images, tested here.

---

### Decision · Program_Chairs · 2023-01-20

**Decision:**

Reject

**Justification For Why Not Higher Score:**

Clarity issue and the baseline issue described above.

**Justification For Why Not Lower Score:**

Can't be lower.

**Metareview: Summary, Strengths And Weaknesses:**

This paper presents an approach for predicting the road networks directly as a graph structure from satellite images.  The approach predicts the graph structure sequentially and reinforcement learning is used to learn the model with MuZero style search.

This paper does not yet meet the bar of acceptance for mainly 2 reasons:
1. Presentation of the technical details of the method is not yet sufficient.  Even though the source code is provided, the authors should still have a clear technical presentation of their approach in the main paper with sufficient details for a knowledgeable reader to be able to reproduce the main idea of this work.  In its current form the reviewers agree that the clarity of the main paper is not yet sufficient to reach this requirement, as in particular the details of RL and the MuZero-style setup was not sufficiently described.
2. The presented results and baselines are a bit questionable.  In particular, the selected baselines for comparison and in particular the base setup used to build up this work seems to be a bit bad.  As reviewer d6d1 pointed out, a base setup only gets 0.455 ALPS which is far worse than any baselines.  This raises the question of why not build upon a better base model?  The proposed method seems to have sufficient potential and general enough to work on any base model, so the authors are strongly encouraged to consider this, which should also push the performance of this system even higher.

**Summary Of Ac-Reviewer Meeting:**

A discussion was held with the AC and all 4 reviewers.  We discussed mostly 3 questions:

First, the usefulness of the problem of road network prediction from satellite images.  One reviewer had questions about this.  After discussion we agreed that this problem is significant and in particular useful in areas where digital data about roads are not readily available.

Second, the issue of presentation clarity.  One reviewer in particular raised the point that the details included in the main paper about RL and MuZero are not sufficient for fully understanding the approach.  The point about the availability of the source code was raised in the discussion, but the reviewers agreed that as a technical paper clarity and completeness of the paper itself should still be sufficiently addressed.

Lastly we discussed the results and the baselines.  Several reviewers realized that the baselines selected for comparison were not state-of-the-art, but decided this hurts the significance a bit but not a big concern.  A more important issue was about the base setup that the authors selected to build their approach upon, which seems to be far worse than the other baselines, therefore a few questions were raised.  In particular what happens if the base setup is reasonably good?  Would the proposed approach not work / do even better?  The reviewers decided that the authors should be encouraged to explore this a bit more and it should also improve the paper.

Overall the consensus was to reject this paper but encourage the authors to improve it further along the above directions.